# DriveArena: A Closed-loop Generative Simulation Platform for Autonomous Driving

## Abstract

This paper introduces DriveArena, the first high-fidelity closed-loop simulation system designed for driving agents navigating real-world scenarios. DriveArena comprises two core components: Traffic Manager, a traffic simulator capable of generating realistic traffic flow on any global street map, and World Dreamer, a high-fidelity conditional generative model with infinite autoregression. DriveArena supports closed-loop simulation using road networks from cities worldwide, enabling the generation of diverse traffic scenarios with varying styles. This powerful synergy empowers any driving agent capable of processing real-world images to navigate in DriveArena's simulated environment. Furthermore, DriveArena features a flexible, modular architecture, allowing for multiple implementations of its core components and driving agents. Serving as a highly realistic *arena* for these *players*, our work provides a valuable platform for developing and evaluating driving agents across diverse and challenging scenarios. DriveArena takes a significant leap forward in leveraging generative models for driving simulation platforms, opening new avenues for closed-loop evaluation of autonomous driving systems.

Codes of DriveArena are attached to the supplementary material.

Project Page: https://blindpaper.github.io/DriveArena/

## 1 Introduction

Autonomous driving (AD) algorithms have advanced rapidly in recent decades (Ayoub et al., 2019; Chen et al., 2023; Xing et al., 2021; Ma et al., 2023; Yang et al., 2021; Mei et al., 2023c;b;a; 2024b), progressing from modular pipelines (Yin et al., 2021; Guo et al., 2023b; Li et al., 2023d; 2022b) to end-to-end models (Hu et al., 2023b; Ye et al., 2023; Jiang et al., 2023) and knowledge-driven methods (Li et al., 2023c; Wen et al., 2023b; Fu et al., 2024b). Despite demonstrating outstanding performance across various benchmarks, significant challenges persist in evaluating these algorithms on replayed open-loop datasets, obscuring their real-world efficacy. Public datasets (Caesar et al., 2020; 2021; Sun et al., 2020), while offering realistic driving data with authentic sensor inputs and traffic behavior, are inherently biased towards simple straight-ahead scenarios. In such cases, an agent can achieve seemingly good performance by merely maintaining its current state, complicating the assessment of actual driving capabilities in complex situations. Furthermore, the agent's current decision does not affect execution or subsequent decisions in the open-loop evaluation, which prevents it from reflecting cumulative errors in real-world driving scenarios. Additionally, the static nature of recorded datasets, where other vehicles cannot react to the ego vehicle's behavior, further hinders the evaluation of AD algorithms in dynamic, real-world conditions.

As illustrated in Figure 1, we analyze existing AD methods and platforms, revealing that most of them are inadequate for a high-fidelity closed-loop simulation. Ideally, as an aspect of embodied intelligence, agents should be evaluated in a closed-loop environment, where other agents react to the actions of the ego vehicle, and the ego vehicle receives changed sensor input accordingly. However, existing simulation environments either cannot simulate sensor inputs (Wen et al., 2023c; Krajzewicz et al., 2012; Gulino et al., 2024) or have a significant domain gap with the real world (Dosovitskiy et al., 2017; Li et al., 2022a), making it difficult to seamlessly integrate algorithms into the real world, thus posing a huge challenge for closed-loop evaluation. We believe that the simulator should not only closely reflect the visual and physical aspects of the real world, but also promote the continuous

learning and evolution of the model within an exploratory closed-loop system for adapting to diverse complex driving scenarios. To achieve this goal, it is imperative to establish a high-fidelity simulator that complies with physical laws and supports interactive functionalities.

Therefore, we present DRIVEARENA, a pioneering closed-loop simulator based on conditional generative models for training and testing driving agents. Specifically, DRIVEARENA offers a flexible platform that can be integrated with any camera-input driving agent. It adopts a modular design and naturally supports iterative upgrades of each module. DRIVEARENA consists of a Traffic Manager that manages traffic flow and a World Dreamer based on autoregressive generation. Traffic Manager can generate realistic interactive traffic flow on any road network worldwide, while World Dreamer is a high-fidelity conditional generative model with infinite autoregression. The driving agent should make corresponding driving actions based on the images generated by World Dreamer, and feed them back to Traffic Manager to update the status of vehicles in the environment. The new scene layout will be returned to World Dreamer for a new round of simulation. This iterative process realizes the dynamic interaction between the driving agent and the simulation environment. The specific contributions are as follows:

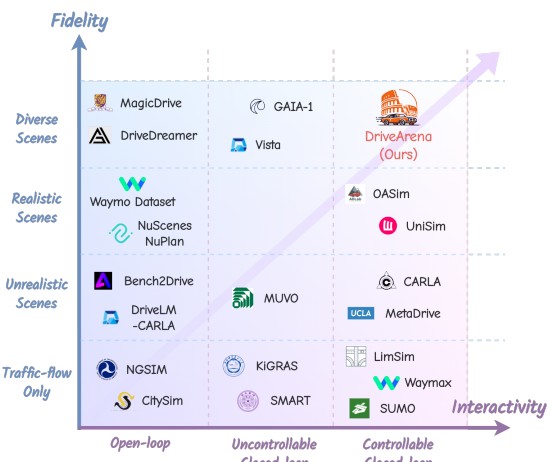

Figure 1: Comparison of DRIVEARENA with existing autonomous driving methods and platforms along the dimensions of Interactivity and Fidelity. *Interactivity* indicates the platform's control over vehicles, *Fidelity* reflects the realism of driving scenarios. DRIVEARENA uniquely occupies the top-right, being the first simulation platform to generate diverse traffic scenarios and surround-view images with closed-loop controllability for all vehicles. For detailed descriptions of these methods and related works, please refer to Table 4 and Appendix A.1.

- **High-fidelity Closed-loop Simulation**: We propose the first high-fidelity closed-loop simulation platform for autonomous driving, DRIVEARENA, which can provide realistic surround images and integrate seamlessly with existing vision-based driving agents. DRIVEARENA closely reflects the visual and physical properties of the real world, enabling agents to continuously learn and evolve in a closed-loop manner and adapt to various complex driving scenarios.

- **Controllability and Scalability**: Our Traffic Manager can dynamically control the movement of all vehicles in the scenarios and feed the road and vehicle layouts into World Dreamer, which utilizes a conditional diffusion framework to generate realistic images in a stable and controllable manner. Additionally, DRIVEARENA supports simulation using road networks from any city worldwide, enabling the creation of diverse driving scenario images with varying styles.

- **Modularized Design**: The Driving Agent, Traffic Manager and World Dreamer communicate via network interfaces, enabling a highly flexible and modular framework. This architecture allows each component to be replaced with different methods without requiring specific implementations. Functioning as an *arena* for these *players*, DRIVEARENA facilitates comprehensive testing and improvement of both vision-based driving agents and driving scene generative models.

## 2 DRIVEARENA FRAMEWORK

As illustrated in Figure 2, the framework of our proposed DRIVEARENA comprises two key components: a Traffic Manager functioning as the backend physical engine and a World Dreamer serving as the real-world image renderer. Notably, DRIVEARENA does not rely on pre-built digital assets or reconstructed 3D road models. Instead, the Traffic Manager adapts to road networks of any city in OpenStreetMap (OSM) format (Haklay & Weber, 2008), which can be directly downloaded from the Internet. This flexibility enables closed-loop traffic simulations on diverse urban layouts.

The Traffic Manager receives ego trajectories output by the autonomous driving agent and manages the movement of all background vehicles. Unlike world model approaches (Gao et al., 2023; Hu

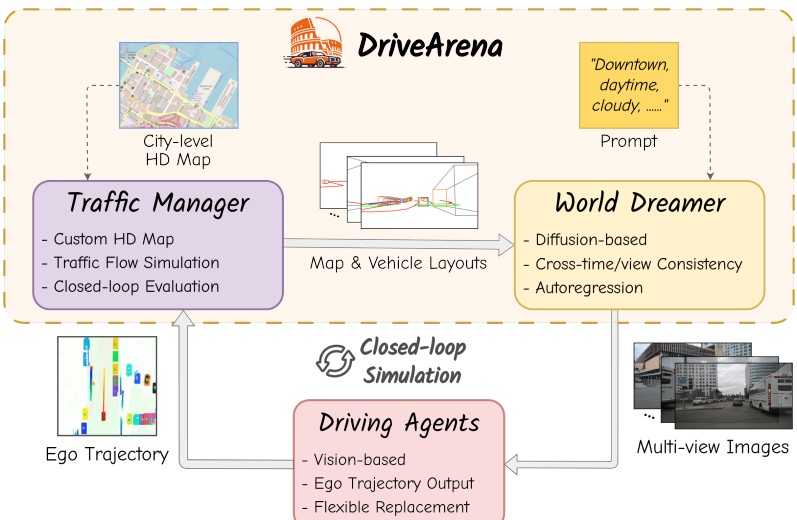

Figure 2: Overview of the DRIVEARENA framework. The system consists of two main components: (1) The Traffic Manager, which processes Internet-downloaded HD maps to create diverse urban layouts, manages vehicle movements including background traffic, and handles collision detection. (2) The World Dreamer, an auto-regressive generative model that generates photo-realistic, multi-view camera images corresponding to the simulation state, with controllable parameters following given prompts. The framework operates in a closed loop: generated images are fed to the AD agent, which outputs the planned ego trajectory. The trajectory is then fed back into the Traffic Manager for the next simulation step.

et al., 2023a) that rely on diffusion models for both image generation and vehicle movement prediction, our Traffic Manager utilizes explicit traffic flow generation algorithms (Wen et al., 2023a). This approach enables the generation of a wider range of uncommon and potentially unsafe traffic scenarios, while also facilitating real-time collision detection between vehicles.

World Dreamer generates realistic camera images that precisely correspond to the Traffic Manager's output. It also allows for user-defined prompts to control various elements of the generated images, such as street view style, time of day, and weather conditions, enhancing the diversity of the generated scenes. Specifically, it employs a diffusion-based model that utilizes the current map and vehicle layouts as control conditions to produce surround-view images. These images serve as input for end-to-end driving agents. Given DRIVEARENA's closed-loop architecture, the diffusion model is required to maintain both cross-view and temporal consistency in the generated images.

The generated multi-view images of the current frame are fed into the end-to-end autonomous driving agents, which can output the ego vehicle's movement. The planned ego trajectory is subsequently sent to DRIVEARENA for the next simulation step. The simulation concludes when the ego vehicle either successfully completes the entire route, crashes, or deviates from the road. Upon completion, DRIVEARENA performs a comprehensive evaluation process to assess the agent's capabilities.

It is noteworthy that DRIVEARENA employs a distributed modular design. The Traffic Manager, World Dreamer, and AD agent communicate via network using standardized interfaces. Consequently, DRIVEARENA does not mandate specific implementations of individual modules and the AD agent. Our framework aims to function as an "*arena*" for these "*players*", facilitating comprehensive testing and improvement of both end-to-end autonomous driving algorithms and realistic driving scene generative models.

## 3 METHODOLOGY

Following the DRIVEARENA framework outlined above, we have implemented a preliminary version of DRIVEARENA. In this section, we elaborate on the implementation of each module: Traffic Manager, World Dreamer, and AD agent, while describing necessary details that were not previously

mentioned. At the end of this section, we present both the open-loop and closed-loop evaluation metrics for AD agents in DRIVEARENA.

## 3.1 TRAFFIC MANAGER

Most existing realistic driving simulators (Yan et al., 2024; Yang et al., 2023b; Wu et al., 2023) rely on limited layouts from public datasets, lacking diversity for dynamic environments. To address these challenges, we utilize LimSim (Wen et al., 2023c; Fu et al., 2024a) as the underlying Traffic Manager to simulate dynamic traffic scenarios and generate road and vehicle layouts for subsequent environment generation. LimSim also provides a user-friendly front-end GUI, which directly displays the BEV map and results from World Dreamer and the driving agent.

Our Traffic Manager enables interactive simulations of multiple vehicles in traffic flow, including comprehensive vehicle planning and control. We adopt a hierarchical multi-vehicle decision-making and planning framework, which jointly makes decisions for all vehicles within the flow and reacts promptly to the dynamic environment through a high-frequency planning module (Wen et al., 2023a). The framework also incorporates a cooperation factor and trajectory weight set, introducing diversity to autonomous vehicles in traffic at both social and individual levels.

Furthermore, our dynamic simulator supports various custom HD maps of any city from Open-StreetMap, facilitating the construction of diverse road graphs for convenient simulation. The Traffic Manager controls the movement of all background vehicles. For the ego vehicle, we provide two distinct simulation modes: open-loop and closed-loop. In closed-loop mode, the driving agent performs planning for the ego vehicle, and Traffic Manager uses the agent-outputted trajectory to control the ego vehicle accordingly. In open-loop mode, the trajectory generated by the driving agent is not actually used to control the ego vehicle; instead, Traffic Manager maintains control in a closed-loop manner. The details of these two modes are further elaborated in Section 3.4.

## 3.2 WORLD DREAMER

Unlike recent autonomous driving generation methods (Yan et al., 2024; Yang et al., 2023b; Wu et al., 2023) that use Neural Radiance Fields (NeRF) and 3D Gaussian Splatting (3D GS) for environment reconstruction from logged video, we design a diffusion-based World Dreamer. It utilizes control conditions of the map and object layouts from the Traffic Manager to generate geometrically and contextually accurate driving scenarios. Our framework shares several advantages: (1) Better controllability. The generated scenes can be controlled by scene layouts from Traffic Manager, textual prompts, and reference images to capture different weather conditions, lighting, and scene styles. (2) Better scalability. Our framework can be adapted to various road structures without the need to model the scene in advance. In theory, we support the generation for any city using OpenStreetMap layouts. However, we acknowledge that compared to NeRF and 3D GS methods, our WorldDreamer currently exhibits limitations in maintaining strict geometric and semantic consistency due to the absence of explicit 3D model constraints.

We illustrate our diffusion-based World Dreamer in Figure 3. Built upon the stable diffusion pipeline (Blattmann et al., 2023), World Dreamer utilizes an effective condition encoding module that accepts a variety of conditional inputs including map and object layouts, text descriptions, camera parameters, ego poses, and reference images to generate realistic surround-view images. Considering the importance of ensuring synthesis scene consistency across different views and time spans for driving agents, we integrate a cross-view attention module, inspired by (Gao et al., 2023), to maintain coherence across different views. Additionally, we adopt an image auto-regressive generation paradigm to enforce temporal consistency. This approach enables World Dreamer to not only maximally maintain the temporal consistency of the generated videos, but also generate videos of arbitrary length in an infinite stream, which provides great support for autonomous driving simulation.

**Condition encoding.** Previous work (Gao et al., 2023) applied the BEV layout as a conditional input to control the output of the diffusion model, increasing the difficulty of the network in learning to generate geometrically and contextually accurate driving scenes. In this work, we introduce more guidance information that helps the model generate high-fidelity surround images. In addition to encoding camera poses for each view, text descriptions, 3D object bounding boxes, and BEV maps using a condition encoder similar to (Gao et al., 2023), we also explicitly project

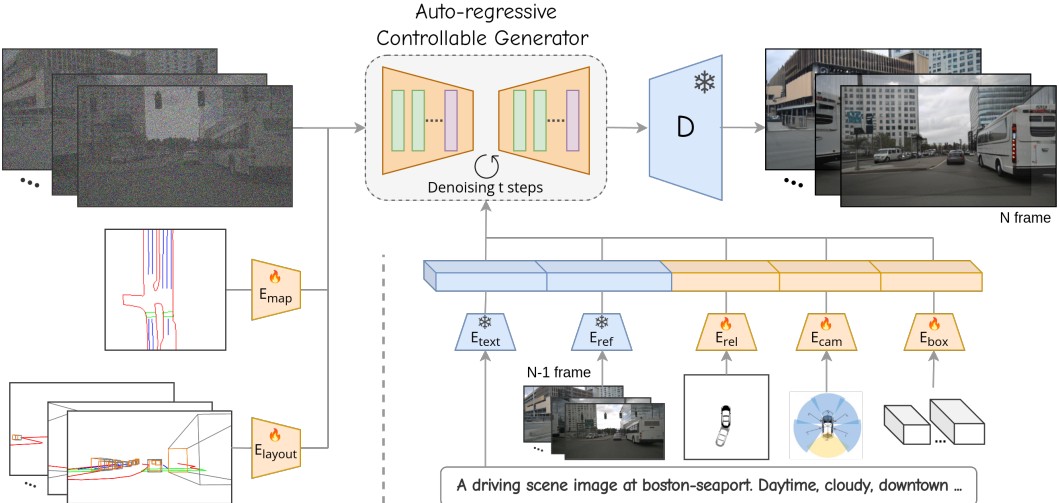

Figure 3: The figure illustrates the denoising process employed by World Dreamer. Beginning with randomly sampled noise, the autoregressive model utilizes various conditions—such as multi-view layout, BEV map, text prompt, reference image, relative pose, camera parameters, and 3D bounding boxes—to enhance the denoising procedure. The encoders depicted in the figure are distinct, with the color indicating whether each one utilizes a pre-trained network or is frozen. Additionally, we incorporate ControlNet to introduce conditional control into the diffusion model.

the map and object layouts onto each camera view to generate layout canvases for more accurate lane and object generation guidance. Specifically, the text embedding $e_{text}$ is obtained by encoding the text descriptions with the CLIP text encoder (Radford et al., 2021). The parameters $\mathbf{P} = \{\mathbf{K} \in \mathbb{R}^{3 \times 3}, \mathbf{R} \in \mathbb{R}^{3 \times 3}, \mathbf{T} \in \mathbb{R}^{3 \times 1}\}$ of each camera and the 8 vertices of the 3D bounding boxes are encoded to $e_{cam}$ and $e_{box}$ by Fourier embedding (Mildenhall et al., 2021), where $\mathbf{K}$, $\mathbf{R}$, $\mathbf{T}$ represent camera intrinsic, rotations and translations respectively. The 2D BEV map grid uses the same encoding method as in (Gao et al., 2023) to obtain the embedding $e_{map}$. Then, each category of the HD maps and the 3D boxes is projected onto the image plane respectively to obtain the layout canvas. The final feature $e_{layout}$ can be obtained by encoding the layout canvas by the conditional encoding network (Zhang et al., 2023).

Moreover, we introduce a reference condition to provide appearance and temporal consistency guidance. During training, we randomly extract a frame from the past n frames as a reference frame and use the pre-trained CLIP model (Radford et al., 2021) to extract reference features $e_{ref}$ from the multi-view images. These features imply semantic context and are integrated into the conditional encoder through a cross-attention module. To enable the diffusion model to grasp how the ego-car's motion influences background changes, we encode the ego-pose relative to the reference frame within the conditional encoder. The relative pose embedding $e_{rel}$ is encoded by Fourier embedding. By incorporating the above control conditions, we can effectively control the generation of images.

**Auto-regressive generation.** To facilitate online inference and streaming video generation while maintaining temporal coherence, we have developed an auto-regressive generation pipeline. Specifically, during the inference phase, the previously generated images and the corresponding relative ego pose are used as reference conditions. This approach guides the diffusion model to generate current surround images with enhanced consistency, ensuring a smoother transition and coherence with the previously generated frames.

This paper presents a simple implementation of World Dreamer. We also verify that extending to a multi-frame auto-regressive version (using multiple past frames as reference and outputting multi-frame images) and adding additional temporal modules can enhance temporal consistency.

## 3.3 DRIVING AGENT

Recent works (Li et al., 2024; Zhai et al., 2023) have demonstrated the challenges in justifying the planning behavior of driving agents through open-loop evaluation on public datasets (Caesar et al.,

2020), primarily due to the simplistic nature of driving scenarios presented. While some studies (Wang et al., 2023a) have conducted closed-loop evaluations using simulators like CARLA (Dosovitskiy et al., 2017), discrepancies such as appearance and scene diversity persist between these simulations and the dynamic real world. To bridge this gap, our DRIVEARENA provides a realistic simulation platform with the corresponding interfaces for camera-based driving agents (Jiang et al., 2023; Hu et al., 2023b; 2022) to perform more comprehensive evaluations, including both open-loop and closed-loop testing. Moreover, by changing the input conditions, such as the road and vehicle layouts, DRIVEARENA could generate corner cases and facilitate these driving agents' evaluation on out-of-distribution scenarios. Without loss of generality, we select two representative end-to-end driving agents, namely UniAD (Hu et al., 2023b) and VAD Jiang et al. (2023), for open-loop and closed-loop testing in our DRIVEARENA. They utilize surround images to predict motion trajectories for the ego vehicle and other agent vehicles, which can be seamlessly integrated with our Traffic Manager for evaluation. Furthermore, the perceptual outputs, such as 3D detection and map segmentation, contribute to enhancing the validation of realism in our environment generation.

## 3.4 EGO CONTROL MODES AND EVALUATION METRICS

DRIVEARENA inherently supports "closed-loop" simulation mode of driving agents. That is, the system adopts the trajectory output by the agent at each timestep, updates the ego vehicle's state based on this trajectory, and simulates the actions of background vehicles. Subsequently, it generates multi-view images for the next timestep, thus maintaining a continuous feedback closed-loop. Additionally, recognizing that some AD agents may be unable to perform long-term closed-loop simulation during the development process, DRIVEARENA also supports the "open-loop" simulation mode. In this mode, the Traffic Manager will take over the control of the ego vehicle, while the trajectory output by the AD agent is recorded for subsequent evaluation.

In both open-loop and closed-loop modes, it is crucial to comprehensively evaluate AD agent performance from a results-oriented perspective. Drawing inspiration from NAVSIM (Dauner et al., 2024) and the CARLA Autonomous Driving Leaderboard (CARLA Team et al., 2023), DRIVEARENA adopts two evaluation metrics: PDM Score (PDMS) and Arena Driving Score (ADS).

PDMS, initially proposed by NAVSIM (Dauner et al., 2024), evaluates the trajectory output at each timestep. We adhere to the original definition of PDMS, which aggregates the following sub-scores:

$$\text{PDMS}_t = \underbrace{\left( \prod_{m \in \{\text{NC}, \text{DAC}\}} \text{score}_m \right)}_{\text{penalties}} \times \underbrace{\left( \frac{\sum_{w \in \{\text{EP}, \text{TTC}, \text{C}\}} \text{weight}_w \times \text{score}_w}{\sum_{w \in \{\text{EP}, \text{TTC}, \text{C}\}} \text{weight}_w} \right)}_{\text{weighted average}}. \tag{1}$$

where the penalties include the drive with no collisions (NC) with road users and drivable area compliance (DAC), as well as the weighted average, including ego progress (EP), time-to-collision (TTC), and comfort (C). We implement minor modifications tailored to DRIVEARENA: in score $_{\text{NC}}$, we do not differentiate "at-fault" collisions, and for score $_{\text{EP}}$, we utilize the Traffic Manager's Ego path planner as the reference trajectory instead of the Predictive Driver Model. At the end of the simulation, the final PDM Score is averaged across all simulation frames.

$$\text{PDMS} = \frac{\Sigma_{t=0}^{T} \text{PDMS}_t}{T} \in [0, 1] \tag{2}$$

For open-loop simulations, PDMS serves directly as the evaluation metric for AD agents. However, for agents operating under the "closed-loop" simulation mode, we employ a more comprehensive metric called Arena Driving Score (ADS), which combines the PDMS with route completion:

$$\text{ADS} = \text{R}_c \times \text{PDMS} \tag{3}$$

where $\text{R}_c \in [0, 1]$ represents route completion, defined as the percentage of the route distance completed by an agent. Given that "closed-loop" simulations terminate upon agent collision with other road users or deviation from the road, ADS provides a suitable metric for differentiating agents' driving safety and consistency.

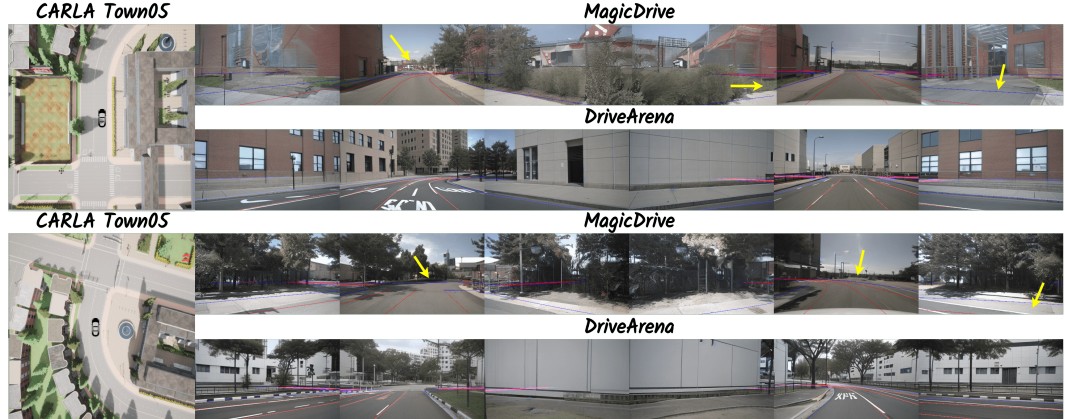

Figure 4: Comparison between MagicDrive and DRIVEARENA. Both are used to generate realistic images on the same Carla Town05 Map, with corresponding ground truth lane lines projected onto the images for demonstration. For such large curvatures and wide roads in CARLA, which are atypical scenarios in the nuScenes dataset, MagicDrive struggles to generate images that accurately fit the network. It incorrectly generates pavements and fails to match the road curvature (indicated by yellow arrows). In contrast, DRIVEARENA successfully generates images that accurately represent the road structure.

## 4 EXPERIMENTS

### 4.1 EXPERIMENTAL SETUPS

For World Dreamer, we use the nuScenes (Caesar et al., 2020) dataset for training. The nuScenes dataset contains data collected from four different cities, covering various lighting and weather conditions, allowing DRIVEARENA to conditionally imitate different appearances. The model is initialized using the pre-trained Stable Diffusion v1.5 (Rombach et al., 2022), and various control conditions are integrated into UNet with a randomly initialized ControlNet (Zhang et al., 2023) to control the denoising process. The experiment is conducted on 8 NVIDIA A100 (80GB) GPUs with a batch size of 4×8 and 200K training iterations.

Traffic Manager operates at 10Hz, while the control frequency is set to 2Hz to accommodate the AD agents. We implement two simulation modes: open-loop and closed-loop. In closed-loop mode, the simulation terminates if it crashes with other vehicles or leaves the road. Currently, DRIVEARENA supports four different maps: `singapore-onenorth`, `boston-seaport`, `boston-thomaspark`, and `carla-town05`. In fact, Traffic Manager can download road network data for any area directly from OpenStreetMap[1] and perform simulations, enabling DRIVEARENA to simulate the road network of almost any city worldwide. Please refer to Appendix A.2 for more implementation details.

### 4.2 WORLD DREAMER PERFORMANCE ASSESSMENT

**Fidelity.** In this section, we assess the sim-to-real gap between our generated images and the original nuScenes images. We generate videos based on the original layout provided by the nuScenes validation set with 2Hz. For comparative analysis, we set MagicDrive as the baseline method perform the same inference using its official code and checkpoints. Subsequently, UniAD is performed as an evaluator on these images to compute various metrics, including 3d object detection, BEV map segmentation, and planning. The results are summarized in Table 1. It shows that all our indicators are higher than the baseline method, and a few indicators even surpass the performance on the original nuScenes. Furthermore, it demonstrates our model's superior capability to accurately respond to control signals and strictly adhere to input conditions. These findings establish a solid foundation for using our generator as a reliable simulator.

---

[1]https://www.openstreetmap.org/

Table 1: Comparison of generation fidelity. The data synthesis conditions are from the nuScenes validation set. All results are computed by using the official implementation and checkpoints of UniAD. **Bold** represents the best results, underline represents the second best results.

| Data Source | 3DOD | | BEV Segmentation mIoU (%) | | | | L2 (m) ↓ | | | | Col. Rate (%) ↓ | | | |
|---|---|---|---|---|---|---|---|---|---|---|---|---|---|---|
| | mAP ↑ | NDS ↑ | Lanes ↑ | Drivable ↑ | Divider ↑ | Crossing ↑ | 1.0s | 2.0s | 3.0s | Avg. | 1.0s | 2.0s | 3.0s | Avg. |
| ori nuScenes | **37.98** | **49.85** | **31.31** | **69.14** | **25.93** | **14.36** | **0.51** | **0.98** | **1.65** | **1.05** | 0.10 | **0.15** | 0.61 | 0.29 |
| MagicDrive | 12.92 | 28.36 | 21.95 | 51.46 | 17.10 | 5.25 | 0.57 | 1.14 | 1.95 | 1.22 | 0.10 | 0.25 | 0.70 | 0.35 |
| DRIVEARENA | 16.06 | 30.03 | 26.14 | 59.37 | 20.79 | 8.92 | 0.56 | 1.10 | 1.89 | 1.18 | **0.02** | 0.18 | **0.53** | **0.24** |

Figure 5: Demonstration of diverse prompts and reference images' influence on identical scenes. The figure features two distinct image sequences generated by DRIVEARENA for the same 30-second simulation, each driven by different prompts and reference images. These sequences reveal clear contrasts in weather and lighting conditions while maintaining their individual styles consistently throughout the entire 30-second duration. For more driving scenes under different prompts and reference images, please refer to Figure 9 in Appendix A.3.

**Controllability and Scalability.** The Traffic Manager can accept any map downloaded from OpenStreetMap and seamlessly connect to the Carla road network. Combined with Dreamer's excellent following capability, the entire framework demonstrates robust controllability and scalability. The specific results are shown in Figure 4. We used both MagicDrive and our World Dreamer to generate realistic images on the same Carla road network, with the corresponding lane lines projected onto the images. The road style in Carla differs significantly from that of nuScenes. It is rare to encounter roads with such large curvature and such wide roads in nuScenes. Consequently, the performance of MagicDrive, which is based on the nuScenes BEV map, is slightly inferior in these conditions. As indicated by the yellow arrow, MagicDrive struggles to generate curved roads and fit wide roads accurately. DRIVEARENA, however, can produce reasonable pictures that follow the road structure.

Figure 5 presents images generated using different text prompts and reference images on the same road network. Each set of images portrays the surrounding scenery at intervals of 8.5 seconds and 24 seconds respectively, with the layout projected on the image. The images clearly illustrate that the road structure and vehicles strictly adhere to the given control conditions while maintaining excellent consistency in the surrounding view. More examples are presented and discussed in Appendix A.3.

Furthermore, the road and vehicle layouts adhere to novel out-of-distribution scenario generation methods. Consequently, ARENA can generate corner cases such as head-on collisions and rear-end collisions. For a detailed elaboration on this aspect, please refer to Appendix A.5.

### 4.3 OPEN-LOOP AND CLOSED-LOOP EXPERIMENTS

In this section, we adopt the prevailing end-to-end autonomous driving methods UniAD (Hu et al., 2023b) and VAD (Jiang et al., 2023) as the driving agents to test both the open-loop and closed-loop performance within the DRIVEARENA framework. We utilized the open-source code and pre-trained

Table 2: Performance of driving agents in DRIVEARENA's open-loop mode. Evaluation across three scenarios: *1)* original nuScenes images sequences; *2)* World Dreamer-generated images with nuScenes ground truth trajectories; and *3)* DRIVEARENA's open-loop mode simulation sequences. Metrics include: no collisions (NC), drivable area compliance (DAC), ego progress (EP), time-to-collision (TTC), comfort (C), and PDM Score (PDMS).

| Scenario | Driving Agent | NC ↑ | DAC ↑ | EP ↑ | TTC ↑ | C ↑ | PDMS ↑ |
|---|---|---|---|---|---|---|---|
| nuScenes (original) | VAD | 0.915±0.17 | 0.937±0.10 | 0.762±0.18 | 0.848±0.23 | **1.000±0.00** | 0.740±0.18 |
| | UniAD | **0.993±0.03** | **0.995±0.01** | **0.914±0.05** | **0.947±0.14** | 0.848±0.21 | **0.910±0.09** |
| nuScenes (generated) | VAD | 0.915±0.16 | 0.942±0.10 | 0.754±0.18 | 0.855±0.23 | **1.000±0.00** | 0.744±0.18 |
| | UniAD | **0.993±0.02** | **0.991±0.02** | **0.909±0.05** | **0.951±0.14** | 0.821±0.21 | **0.902±0.09** |
| DRIVEARENA | VAD | **0.807±0.11** | **0.950±0.05** | **0.795±0.13** | **0.800±0.12** | **0.913±0.09** | **0.683±0.12** |
| | UniAD | 0.792±0.11 | 0.942±0.04 | 0.738±0.11 | 0.771±0.12 | 0.749±0.16 | 0.636±0.08 |
| nuScenes GT | Human | 1.000±0.00 | 1.000±0.00 | 1.000±0.00 | 0.979±0.12 | 0.752±0.17 | 0.950±0.06 |

weights from the two driving agents without additional training. UniAD and VAD operate at 2Hz, outputting a trajectory of 6 path points over the next 3 seconds. Traffic Manager further interpolates this to a 10Hz trajectory.

**Open-loop Evaluation.** We first assess the performance of driving agents in DRIVEARENA's open-loop mode. The agents are evaluated on three scenarios: *1)* the original nuScenes image sequences; *2)* World-Dreamer-generated nuScenes image sequences, where the vehicles' trajectory remains identical to nuScenes ground truth, but surround images are replaced with World-Dreamer-generated ones; and *3)* DRIVEARENA's own simulation sequences (i.e., DRIVEARENA's open-loop mode). Our evaluation metrics consist of the PDM Scores and its sub-scores, as detailed in Section 3.4. Additionally, we evaluate trajectories driven by human drivers in nuScenes as the human driver performance. Detailed results are presented in Table 2.

The results show that UniAD performs best on the original nuScenes sequence with a PDMS metric of 0.91, whereas the PDMS metric on the World Dreamer-generated sequence is a surprising 0.902, with a metric drop of less than 1%. We attribute this to both the high fidelity of our World Dreamer-generated images and UniAD's dependence on ego states, as corroborated by (Li et al., 2024). Furthermore, VAD achieves better performance on the World Dreamer simulation sequences with a PDMS of 0.683, demonstrating its better open-loop ability on unseen roadmaps and scenarios. Figure 6 presents the open-loop performance of both driving agents in long simulation sequences. More examples of visualization of the open-loop evaluation can be found in Appendix A.4.

**Closed-loop Evaluation.** We further evaluate the performance of VAD and UniAD in DRIVEARENA's closed-loop mode. In this mode, the trajectory outputted by the driving agents is directly used for ego vehicle control, and the evaluation metrics include PDM Score (PDMS), Route Completion (RC), and Arena Drive Score (ADS). Our closed-loop experiment was conducted on four pre-set paths, with two paths selected in Boston and two in Singapore. The simulation time to complete each route was approximately 120 seconds. Detailed results are presented in Table 3.

The results indicate that the PDMS of UniAD-generated trajectories in closed-loop mode (Avg.: 0.667) are comparable to those of the open-loop mode, while the PDMS of VAD-generated trajectories (Avg.: 0.534) has a significant metric drop of 0.149. The Route Completions (RC) of both driving agents are consistently low, with VAD completing just 4.59% and UniAD completing an average of 13.7% of the route. During the evaluation, both agents performed better on straightaways but largely failed to navigate the first turning intersection

Table 3: Evaluation of Driving Agents' performance in DRIVEARENA's closed-loop mode across four distinct routes. Performance metrics include: PDM Score (PDMS), Route Completion (RC), and Arena Driving Score (ADS).

| Route | Driving Agent | PDMS ↑ | RC ↑ | ADS ↑ |
|---|---|---|---|---|
| sing_route_1 | VAD | 0.5315 | 0.0467 | 0.0248 |
| | UniAD | **0.7615** | **0.1684** | **0.1282** |
| sing_route_2 | VAD | 0.5147 | 0.0400 | 0.0206 |
| | UniAD | **0.7215** | **0.169** | **0.0875** |
| bos_route_1 | VAD | **0.5830** | 0.0604 | 0.0352 |
| | UniAD | 0.4952 | **0.091** | **0.0450** |
| bos_route_2 | VAD | 0.5054 | 0.0366 | 0.0185 |
| | UniAD | **0.6888** | **0.121** | **0.0835** |

in the route. While VAD showed better metrics in open-loop mode, it failed to achieve better results when conducting closed-loop experiments. This highlights the inherent flaws of open-loop

evaluation in assessing the true capabilities of agents, which could potentially be mitigated by DRIVEARENA. See Appendix A.4 for the visualization of the failure cases in DRIVEARENA closed-loop mode.

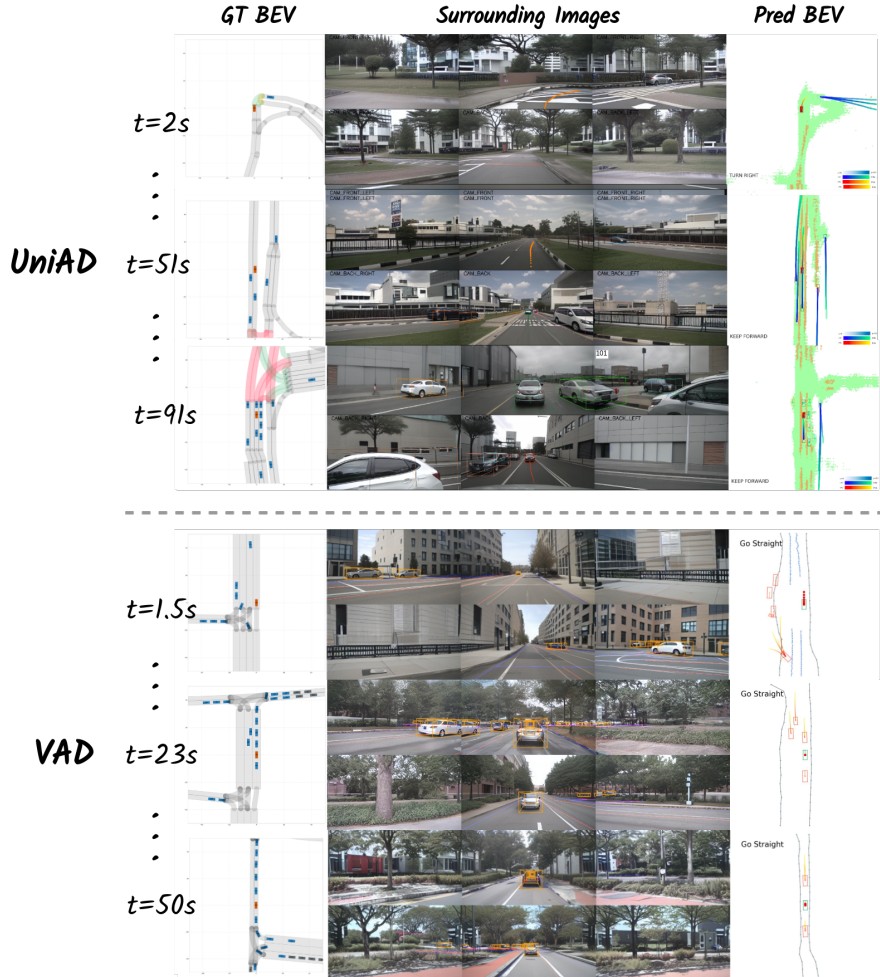

Figure 6: Case studies on open-loop performance of driving agents in DRIVEARENA. Two long-term open-loop simulation sequences are shown: the upper sequence depicts the performance of the uniad on the road network and style (left-hand drive) in Singapore, while the lower sequence shows the performance of VAD on the road network and style (right-hand drive) in Boston. Each subfigure shows, in order from left to right: ground truth BEVs from Traffic Manager; World Dreamer-generated images; and agent-predicted BEV images. For more open-loop case illustrations, please refer to Figure 12 and 13 in Appendix A.4.

## 5 CONCLUSIONS

This paper introduces a novel closed-loop simulation platform named DRIVEARENA for vision-based driving agents. DRIVEARENA integrates a Traffic Manager that generates human-like traffic flow and a high-fidelity generative World Dreamer with infinite generation. This combination allows realistic interaction and continuous feedback between the driving agent and the simulation environment. The system provides a valuable platform for developing and testing autonomous driving agents in a variety of scenarios, marking a substantial leap in driving simulation technology. DRIVEARENA is designed with a modular architecture, allowing for easy replacement of each module. As the first high-fidelity closed-loop simulator, we still have a few limitations for future improvement, which are discussed in detail in Appendix A.6.

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

# A  APPENDIX

CONTENTS

## A.1  RELATED WORKS

Table 4: Comparison of various datasets, generative models, world models, and simulators in terms of interactivity, fidelity, and diversity features. **DATA.** represents dataset, **GEN.** represents generative model, **W.M.** represents world model, **SIM.** represents simulator.

| Type | Name | Interactivity Uncontrollable Closed-loop | Controllable Closed-loop | Fidelity Realistic Images | Real-world Roadgraph | Different daylight/weather | Diversity Multi-view Images | Unlimited Video | Unspecified map |
|---|---|---|---|---|---|---|---|---|---|
| DATA. | CitySim (Robinson et al., 2009) / NGSIM | ✗ | ✗ | ✗ | ✔ | ✔ | ✗ | ✗ | ✗ |
| | Bench2Drive (Jia et al., 2024) | ✗ | ✗ | ✗ | ✗ | ✔ | ✔ | ✗ | ✗ |
| | DriveLM-CARLA (Sima et al., 2023) | ✗ | ✗ | ✗ | ✗ | ✔ | ✔ | ✗ | ✗ |
| | nuPlan dataset (Caesar et al., 2021) | ✗ | ✗ | ✔ | ✔ | ✗ | ✔ | ✗ | ✗ |
| | nuScenes (Caesar et al., 2020) / Waymo dataset (Sun et al., 2020) | ✗ | ✗ | ✔ | ✔ | ✔ | ✔ | ✗ | ✗ |
| GEN. | MagicDrive (Gao et al., 2023) / DriveDreamer (Wang et al., 2023b) | ✗ | ✗ | ✔ | ✔ | ✔ | ✔ | ✗ | ✗ |
| | SimGen (Zhou et al., 2024) | ✗ | ✗ | ✔ | ✔ | ✔ | ✗ | ✗ | ✗ |
| W.M. | KiGRAS (Zhao et al., 2024a) / SMART (Wu et al., 2024) | ✔ | ✗ | ✗ | ✔ | ✗ | ✗ | ✗ | ✔ |
| | MUVO (Bogdoll et al., 2023) | ✔ | ✗ | ✗ | ✔ | ✗ | ✗ | ✗ | ✗ |
| | Vista (Gao et al., 2024) / GAIA-1 (Hu et al., 2023a) | ✔ | ✗ | ✔ | ✗ | ✔ | ✗ | ✗ | ✗ |
| SIM. | Waymax (Gulino et al., 2024) | ✔ | ✔ | ✗ | ✔ | ✗ | ✗ | ✗ | ✗ |
| | SUMO (Krajzewicz et al., 2012) / LimSim (Wen et al., 2023c) | ✔ | ✔ | ✗ | ✔ | ✗ | ✗ | ✗ | ✔ |
| | CARLA (Dosovitskiy et al., 2017) | ✔ | ✔ | ✗ | ✔ | ✔ | ✔ | ✔ | ✔ |
| | MetaDrive (Li et al., 2022a) | ✔ | ✔ | ✗ | ✔ | ✗ | ✔ | ✔ | ✔ |
| | Unisim (Yang et al., 2023b) / OAsim (Yan et al., 2024) | ✔ | ✔ | ✔ | ✔ | ✗ | ✔ | ✗ | ✗ |
| **Ours** | **DRIVEARENA** | ✔ | ✔ | ✔ | ✔ | ✔ | ✔ | ✔ | ✔ |

### A.1.1  DATA ACQUISITION FOR AUTONOMOUS DRIVING

The characteristics of the automated driving dataset can be categorized into two aspects: appearance fidelity and interactivity. First, regarding appearance fidelity, NGSIM (U.S. Department of Transportation Federal Highway Administration, 2016) and CitySim (Robinson et al., 2009) provide only realistic driving trajectories that can provide safe and reliable driving planning guidance. On top of that, some datasets developed based on the Carla simulator, such as DriveLM-CARLA (Sima et al., 2023) and Bench2Drive (Jia et al., 2024), provide simulated sensor data. Taking it a step further, the Waymo (Sun et al., 2020) and nuScenes (Caesar et al., 2020) datasets capture real-world sensor recordings and the driving behavior of human drivers. The datasets were produced in a complex process and with limited data. To add variety to the scenarios, MagicDrive (Gao et al., 2023) and DriveDreamer (Wang et al., 2023b) provide editable scenario generation. So far, we have obtained diverse and rich data for training. However, the above data can only be used for open-loop evaluation, i.e., current decisions do not affect future data distributions, which differs significantly from

actual driving. Works (Hu et al., 2023a; Gao et al., 2024; Bogdoll et al., 2023; Zhao et al., 2024a; Wu et al., 2024) that also have fidelity differences improve the interactivity of the data, they usually use auto-regressive generation methods to realize the interaction, the generation process implies the model's understanding of the world. Usually, it can not be too much human intervention. Some simulators (Gulino et al., 2024; Wen et al., 2023c; Li et al., 2022a; Dosovitskiy et al., 2017; Yang et al., 2023b; Yan et al., 2024; Krajzewicz et al., 2012) make things more controllable by decoupling part of the mechanics of how the world works. Common examples include simulators (Krajzewicz et al., 2012; Gulino et al., 2024; Wen et al., 2023c) that provide realistic traffic flow, and simulators (Dosovitskiy et al., 2017; Li et al., 2022a) that drive vehicles in game engines, and reconstructive simulations represented by (Yang et al., 2023b; Yan et al., 2024) that provide the appearance of reality.

### A.1.2 Diffusion-based Generative Models

Recent advancements in generative models have seen diffusion models play a pivotal role in image and video generation (Dhariwal & Nichol, 2021; Meng et al., 2021; Nichol et al., 2021; Podell et al., 2023; Ramesh et al., 2022; Blattmann et al., 2023; He et al., 2022). Moreover, recent works have expanded the scope by integrating additional control signals beyond traditional text prompts (Guo et al., 2023a; Li et al., 2023e; Mou et al., 2024). For instance, ControlNet (Zhang et al., 2023) incorporates a trainable version of the SD encoder for control signals, while studies such as Uni-ControlNet (Zhao et al., 2024b) and UniControl (Qin et al., 2023) have emphasized the fusion of multi-modal inputs into a unified control condition using input-level adapter structures. Our approach aims to study the generation of continuous and controllable sequence frames, thereby bridging the gap between simulation environments and reality, and establishing the required foundation for closed-loop learning of autonomous driving agents.

### A.1.3 Evolution of Autonomous Driving Generation

World Models (Hu et al., 2023a; Yang et al., 2024) utilize diffusion models to generate future driving scenes based on historical information. These methods often lack the ability to control the scenarios through layout, are difficult to generate continuous and stable videos and lack the approximation of physical laws.

TrackDiffusion focused on generating videos based on 2D object layouts (Li et al., 2023a). BEV-Gen (Swerdlow et al., 2024) pioneered the generation of synthetic multi-view images based on the BEV layout, laying the foundation for a controllable generation of autonomous driving scenarios. BEVControl (Yang et al., 2023a) extended this approach by a height elevation process, enabling image generation aligned with surrounding projection layouts. Further advancements includes MagicDrive (Gao et al., 2023), DriveDreamer (Wang et al., 2023b), Panacea (Wen et al., 2024) and DrivingDiffusion (Li et al., 2023b), which generate panoramic controllable videos through various 3D controls and encoding strategies. However, their primary focus lies in augmenting training data to enhance algorithm performance, rather than serving as simulators for modeling dynamic environmental interactions.

### A.1.4 Simulator-Driven Scenario Generation

Autonomous vehicle development is significantly enhanced by driving simulators, which provide controlled environments for realistic simulation. Prominent research efforts have concentrated on generating virtual imagery and annotations, with some studies expanding to incorporate environmental variations and construct safety-critical scenarios for training based on real-world data logs. Nevertheless, these simulated images frequently fall short of achieving true realism, as evidenced by previous works (Ros et al., 2016; Richter et al., 2016; Sun et al., 2022). While SimGen (Zhou et al., 2024) made a breakthrough as the first work to generate diverse driving scenarios following conditions from a simulated environment, it mainly focused on the quality of the generated content with only front-view images, neglecting the exploration of closed-loop systems. Our research aims to bridge this gap by developing a system that can not only generate realistic scenarios but also allow agents to interact with them in a closed-loop manner.

### A.1.5 CLOSED-LOOP DRIVING IN SIMULATION

End-to-end vehicle control algorithms (Hu et al., 2022; 2023b; Ye et al., 2023), are typically trained and evaluated on open-loop datasets (Caesar et al., 2020). However, these algorithms lack the capability to generalize directly to simulators for closed-loop evaluation, which hinders the demonstration of their true performance potential. Recent studies have increasingly recognized the significance of closed-loop evaluation, as exemplified by (Jiang et al., 2023; Wang et al., 2023a). Moreover, simulation environments offer a wealth of training data, a stark contrast to models trained on datasets that are constrained by data distribution (Li et al., 2024). A significant challenge arises due to the discrepancy between the simulated scene's appearance and real-world conditions, complicating the generalization of models trained on simulation data to actual scenarios. This creates a paradox: the desire to utilize simulation data for its diversity and editability, while also seeking data that closely mirrors reality. Our approach effectively addresses this issue by enhancing the realism of the simulator for certain closed-loop learning methods (Mei et al., 2024a).

## A.2 IMPLEMENTATION DETAILS

### A.2.1 WORLD DREAMER SETUPS

**Dataset.** For World Dreamer, we use the nuScenes (Caesar et al., 2020) dataset for training. Following the official configuration, we employ 700 scenes for training and 150 for validation. We focus on four road categories (lane boundary, lane divider, pedestrian crossing, and drivable area) and ten object categories. The nuScenes dataset contains data collected from four different cities, covering various light and weather conditions, including daytime, night, sunny, cloudy, and rainy scenarios, enabling DRIVEARENA to conditionally imitate diverse appearances. We additionally annotated each scene using GPT-4V, providing detailed scene descriptions that include elements like time, weather, street style, road structure, and appearance. These descriptions serve as text prompt conditions.

**Model Setup.** The model is initialized with the pre-trained Stable Diffusion v1.5 (Rombach et al., 2022), with only the newly added parameters being trained. For various conditions, except for the encoding of reference images and text prompts, the encoders for other conditions are randomly initialized and trained from scratch. These conditions are then integrated into the UNet using a randomly initialized ControlNet (Zhang et al., 2023) to control the denoising process.

**Training and Inference.** To utilize the reference images and achieve temporal correlation, we employ ASAP (Wang et al., 2023c) to generate 12Hz interpolated annotations and crop them into image clips of length n = 7. During training, we use the last frame of each clip as the current frame, select any frame from the clip as the reference frame, and calculate the relative pose between them to model the motion trend of the background. Accordingly, the surround images corresponding to the reference frame are input to the network as reference images. During inference, the generated result of the previous frame is used as the current reference images, enabling unlimited length generation. The experiment is conducted on 8 NVIDIA A100 (80GB) GPUs with a batch size of 4×8 and 200K iterations of training. The AdamW optimizer is used with a learning rate of 1e-4. The network follows the same image resolution (224×400) as MagicDrive, and when input to the driving agent, it will be upsampled to the original image size of nuScenes (900×1600) through a super-resolution algorithm (Wang et al., 2024).

### A.2.2 TRAFFIC MANAGER SETUPS

**Operating Frequencies.** In our experiments, the Traffic Manager operates at a frequency of 10Hz, while the control frequency is set to 2Hz. This configuration results in the Traffic Manager sending the current layout to World Dreamer every 0.5 simulation seconds, requesting surround images. These images are then forwarded to the driving agent, which predicts and plans the subsequent trajectory for the ego vehicle. The Traffic Manager, World Dreamer, and driving agent communicate via HTTP protocol, enabling deployment across different servers.

**Simulation Modes.** As detailed in Section 3.4, we implement two simulation modes. In the open-loop mode, all vehicles, including the ego vehicle, are controlled by Traffic Manager itself. The driving agent can predict the ego vehicle's trajectory, but its trajectory is not actually executed. In the

closed-loop mode, the ego vehicle is controlled by the driving agent, and the simulation terminates if it crashes with other vehicles or leaves the road.

**Supported Maps.** Currently, DRIVEARENA supports four different maps, which are: `singapore-onenorth`, `boston-seaport`, `boston-thomaspark`, and `carla-town05`. The first two maps closely resemble the corresponding areas in the nuScenes dataset, while the last one replicates the road network of the Town05 map in the CARLA simulator. Notably, Traffic Manager can download road network data for any area directly from Open-StreetMap and perform simulations, enabling DRIVEARENA to simulate the road network of almost any city worldwide. Our map processing pipeline employs a two-stage approach using SUMO tools to enhance OSM's road-level information. First, we utilize `OSMWebWizard` to download OSM maps and establish topological roadnet. Second, we employ the `randomTrips` script to generate vehicle demands and their corresponding origin-destination pairs within the map. Beyond these pre-simulation processes, we support several customization options where users can modify downloaded OSM maps, create custom maps manually, or convert OpenDRIVE format maps to our supported format.

### A.3 VISUALIZATION OF WORLD DREAMER

In this section, we will comprehensively demonstrate the controllability and scalability of the model from various dimensions, including the control of lighting and weather, the fit of object boxes and maps, change of street style, and consistency over long periods of time.

We conducted an experiment by setting up two identical traffic scenes sharing the same road network and traffic participants, varying only the ego vehicle's position. The results generated by World Dreamer, shown in Figure 8, demonstrate how the model handles scene consistency when the ego vehicle moves from the leftmost to the middle lane. World Dreamer successfully maintains spatial consistency in lane markings and surrounding vehicle positions while preserving similar street styles and building configurations. However, due to the inherent stochastic nature of diffusion models, minor variations emerge in vehicle colors and street backgrounds.

We demonstrate the impact of the reference image on the generated image, as shown in Figure 7. We randomly select one frame of images from the nuScenes dataset as reference images and choose three scenes from OpenStreetMap and Carla. We perform inference on them with World Dreamer respectively. It can be seen that the source and style of the road network are very different from the scope of the original nuScenes dataset. The pictures show that the generated vehicles and road networks conform closely to control conditions, demonstrating strong control capabilities. The style and weather of the generated pictures can also be consistent with the reference images. In other words, besides maintaining image generation continuity through reference images, we can also regulate image style accordingly.

In addition to the two weather generation examples shown in Figure 5, Figure 9 presents two more demonstrations, further highlighting the controllability and fidelity of World Dreamer. These four sets of images display notable variations in weather and lighting while consistently maintaining their distinct styles throughout the continuous iteration process.

We demonstrate additional cases using data from the nuPlan dataset to validate the scalability. The nuPlan data originates from cities different from nuScenes and features varying camera numbers and parameters. We select 6 cameras with a similar layout to the nuScenes dataset, and nuPlan's camera parameters are employed to project object boxes and lane lines onto corresponding images as control conditions. As shown in Figure 10, World Dreamer which only trained on nuScenes adeptly adheres to these conditions, generating coherent images when deployed in new cities and even with novel camera configurations.

In addition, we trained a version of World Dreamer using a mixed training dataset combining both nuPlan and nuScenes datasets. While maintaining the same model architecture, we found that World Dreamer can generate street view images in the distinctive styles of Las Vegas and Pittsburgh, which are exclusively present in the nuPlan dataset. The results are shown in Figure 11, where we can observe the characteristic Las Vegas Strip with its palm trees, as well as the distinctive low-rise buildings of Pittsburgh. By incorporating more diverse driving data, we successfully enhanced the generative model's capability to generalize across different urban environments.

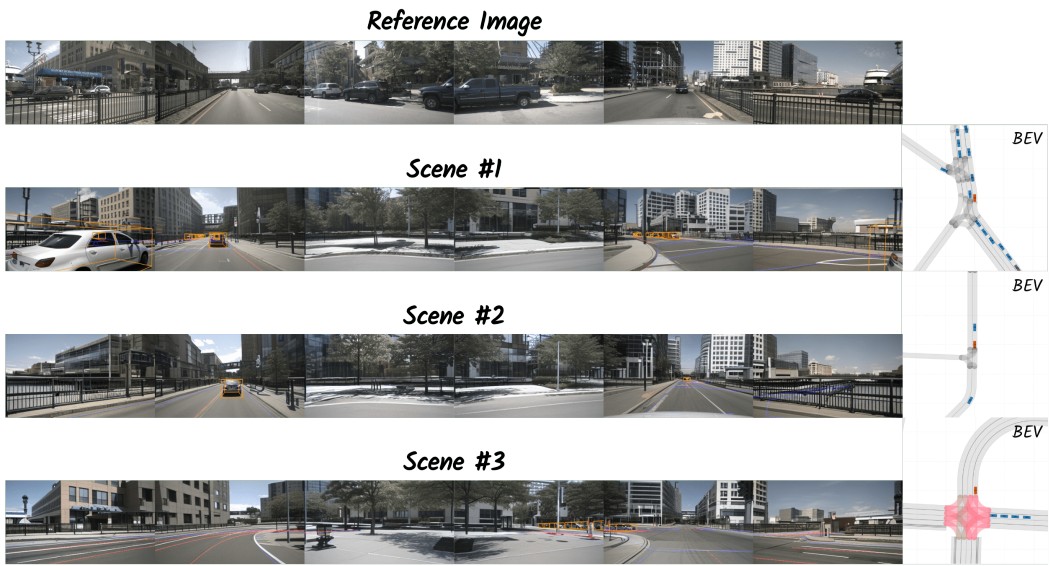

Figure 7: Demonstration of reference image influence on generated scenes. Three scenes are presented, all derived from a single nuScenes reference frame. Despite notable variations in road networks, World Dreamer successfully integrates street styles and weather conditions from the reference image while adhering to specified control conditions for vehicles and road layouts. Of particular interest is the aerial corridor visible in the reference image, which is accurately reproduced in scenes #1 and #2. However, in scene #3, due to the curved road configuration, the corridor is not generated, illustrating World Dreamer's adaptability to different road geometries.

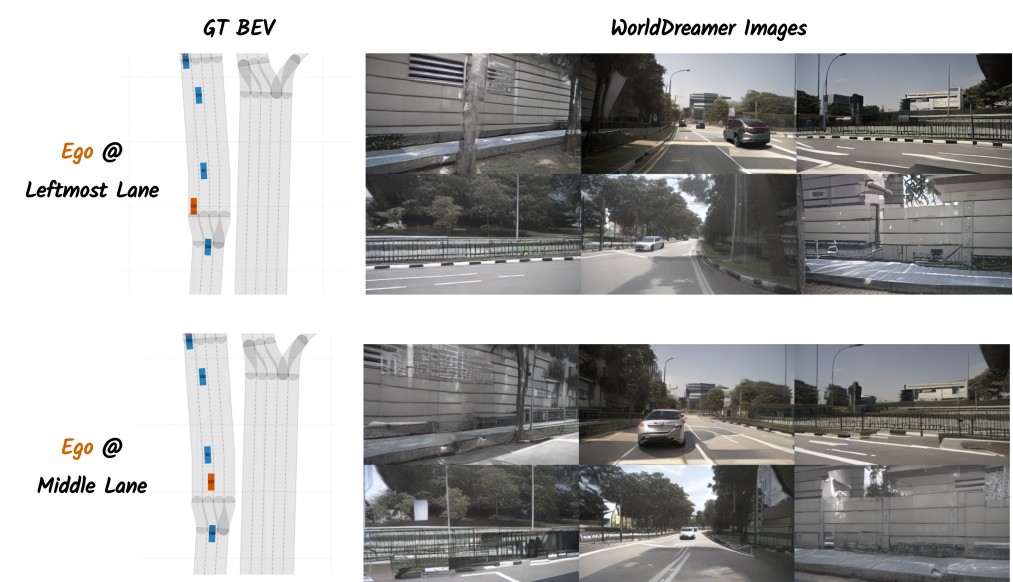

Figure 8: Demonstration of generated images from identical traffic scenarios with varying ego positions. The two scenes share the same road network and traffic participants, with the ego vehicle position shifting from the leftmost to the middle lane. While minor variations appear in front vehicle color and street backgrounds, World Dreamer maintains consistent lane markings and spatial relationships of surrounding vehicles, preserving similar street styles and building configurations.

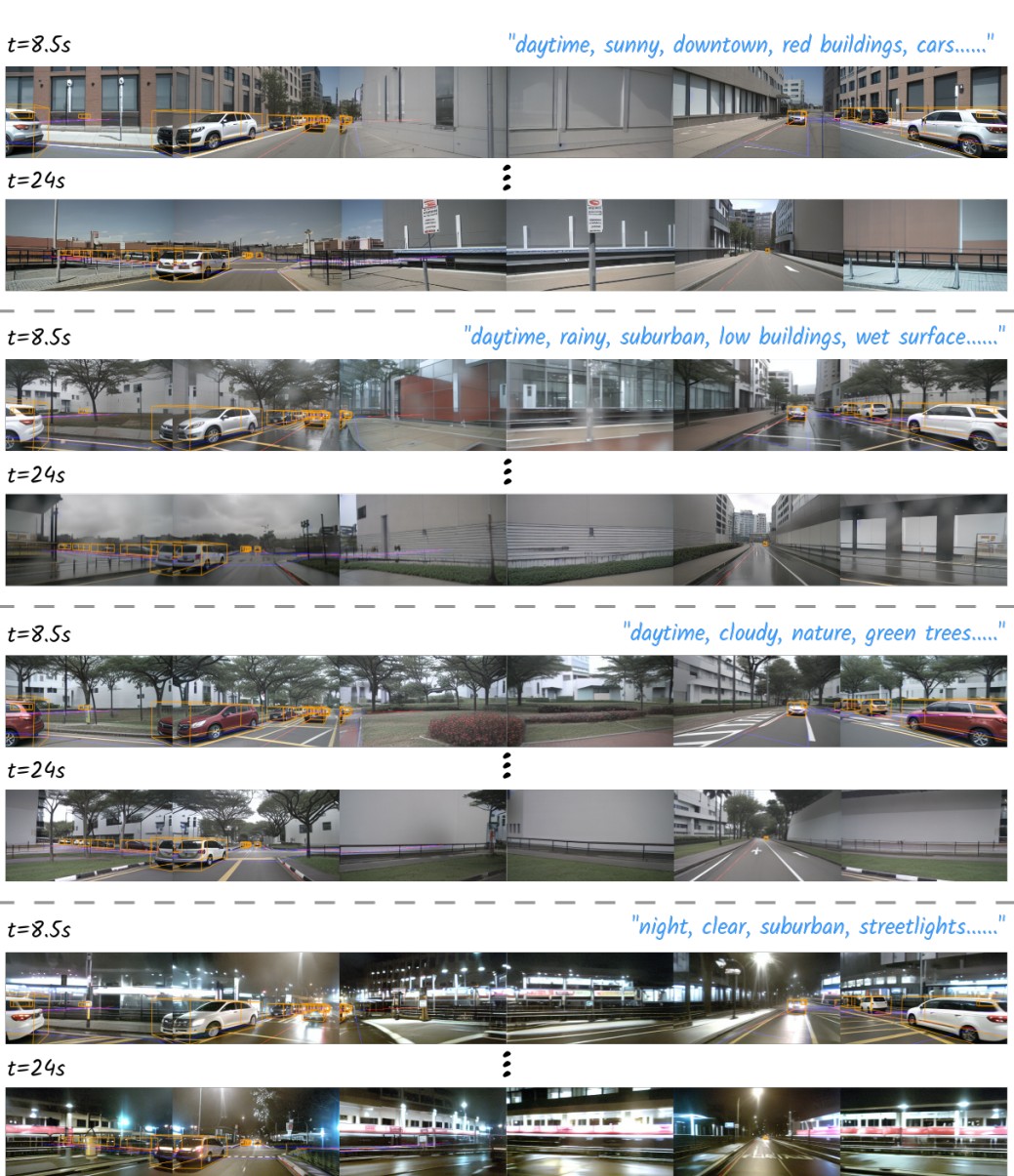

Figure 9: Demonstration of diverse prompts and reference images' influence on identical scenes. The figure presents four distinct image sequences generated by DRIVEARENA for the same 30-second simulation sequence, each utilizing different prompts and reference images. All sequences strictly adhere to the provided control conditions for road structures and vehicles, maintaining cross-view consistency. Notably, the four sequences exhibit significant variations in weather and lighting conditions while consistently preserving their respective styles throughout the entire 30-second duration.

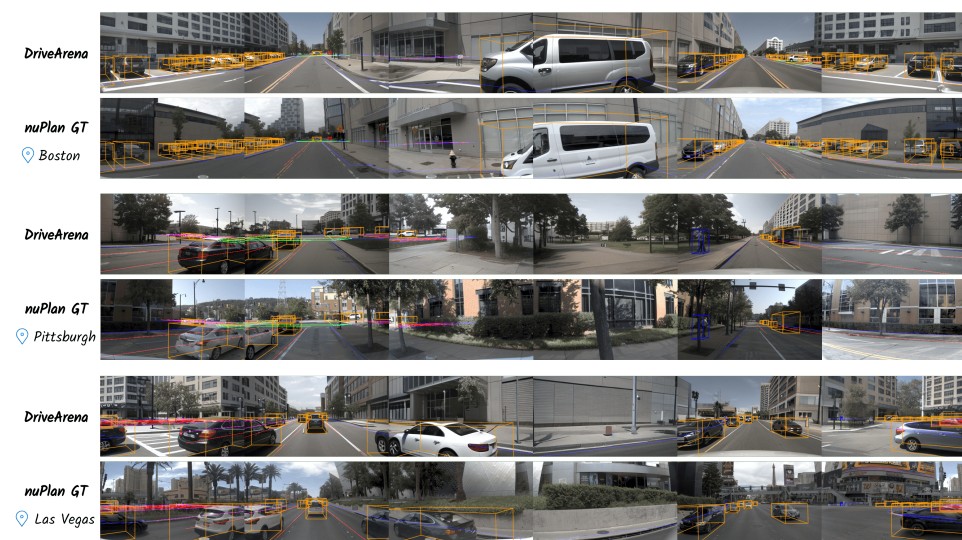

Figure 10: Zero-shot inference on nuPlan datasets. World Dreamer, trained exclusively on the nuScenes dataset, demonstrates remarkable adaptability when applied to the nuPlan dataset. The latter comprises data from new cities (Pittsburgh, Las Vegas) that are not present in nuScenes, with different camera configurations and parameters. We selected three nuPlan scenes and directly utilized nuPlan's camera parameters to project object boxes and lane lines onto the corresponding images as control conditions. The results show that World Dreamer produces coherent images when deployed in unfamiliar cities and even with previously unseen camera configurations and layouts.

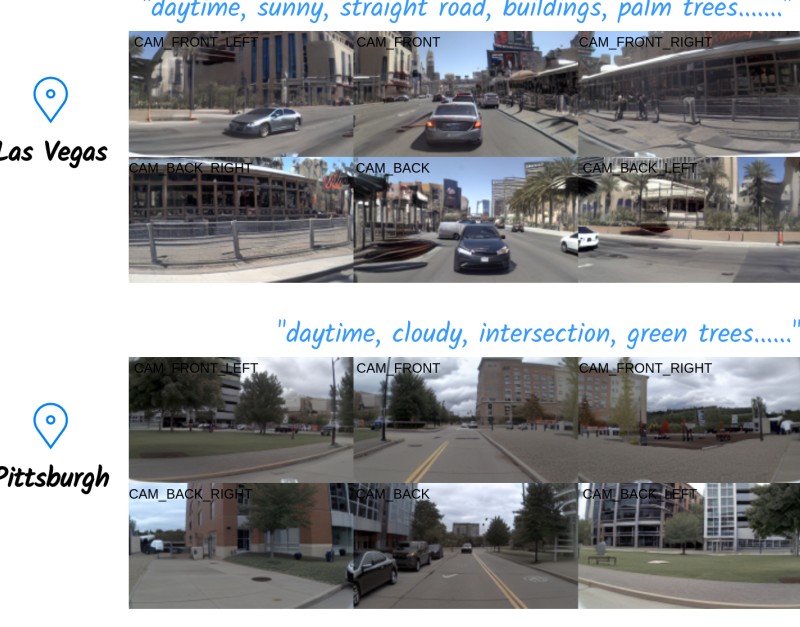

Figure 11: World Dreamer inference on nuPlan locations. Using a mixed training dataset from both nuPlan and nuScenes, World Dreamer demonstrates the ability to generate street view images capturing the distinctive styles of Las Vegas and Pittsburgh—locations exclusive to the nuPlan dataset. This enhanced training approach with diverse driving data significantly improves World Dreamer's generalization capability.

## A.4 VISUALIZATION OF OPEN-LOOP AND CLOSE-LOOP EXPERIMENTS

In DRIVEARENA's open-loop mode, Figure 12 and Figure 13 illustrate two additional sequences on top of Figure 6, demonstrating that the prediction from the driving agents on the road network and vehicle tracking is fundamentally accurate. However, in terms of metrics, performance from both agents in such scenarios with unseen road and traffic flow is significantly degraded, with an average PDM Score of only 0.636 for UniAD and an average PDM Score of 0.683 for VAD. The output trajectories exhibit a substantial increase in collision rates and instances of driving outside the drivable area.

Figure 14 illustrates two failure cases where UniAD lacked sufficient trajectory correction capabilities. Despite a roughly correct prediction of the road structure, it ultimately mounted the central green belt or failed to complete a right turn successfully. The average Arena Driving Score for UniAD is 0.086, while VAD achieves only 0.025 on average. Two failure cases resulting from the VAD's closed-loop evaluation in DRIVEARENA are presented in Figure 15. In failure case 1, the driving agent ran onto the central tree lawn while recognizing the right road boundary comparatively correctly. In Failure Case 2, the VAD incorrectly predicted the left-turn roadway structure as a straight roadway and, therefore, could not successfully complete the left turn. These cases demonstrate the importance of closed-loop evaluation in reflecting the true capabilities of AD agents, and also show that our DRIVEARENA demonstrates good ability in following the road structure.

It should be noted that these are preliminary results based on testing only 4 routes. We plan to expand the number of routes for a more comprehensive evaluation and explore the combined effect of World Dreamer's timing consistency and the driver agent's performance on the final ADS.

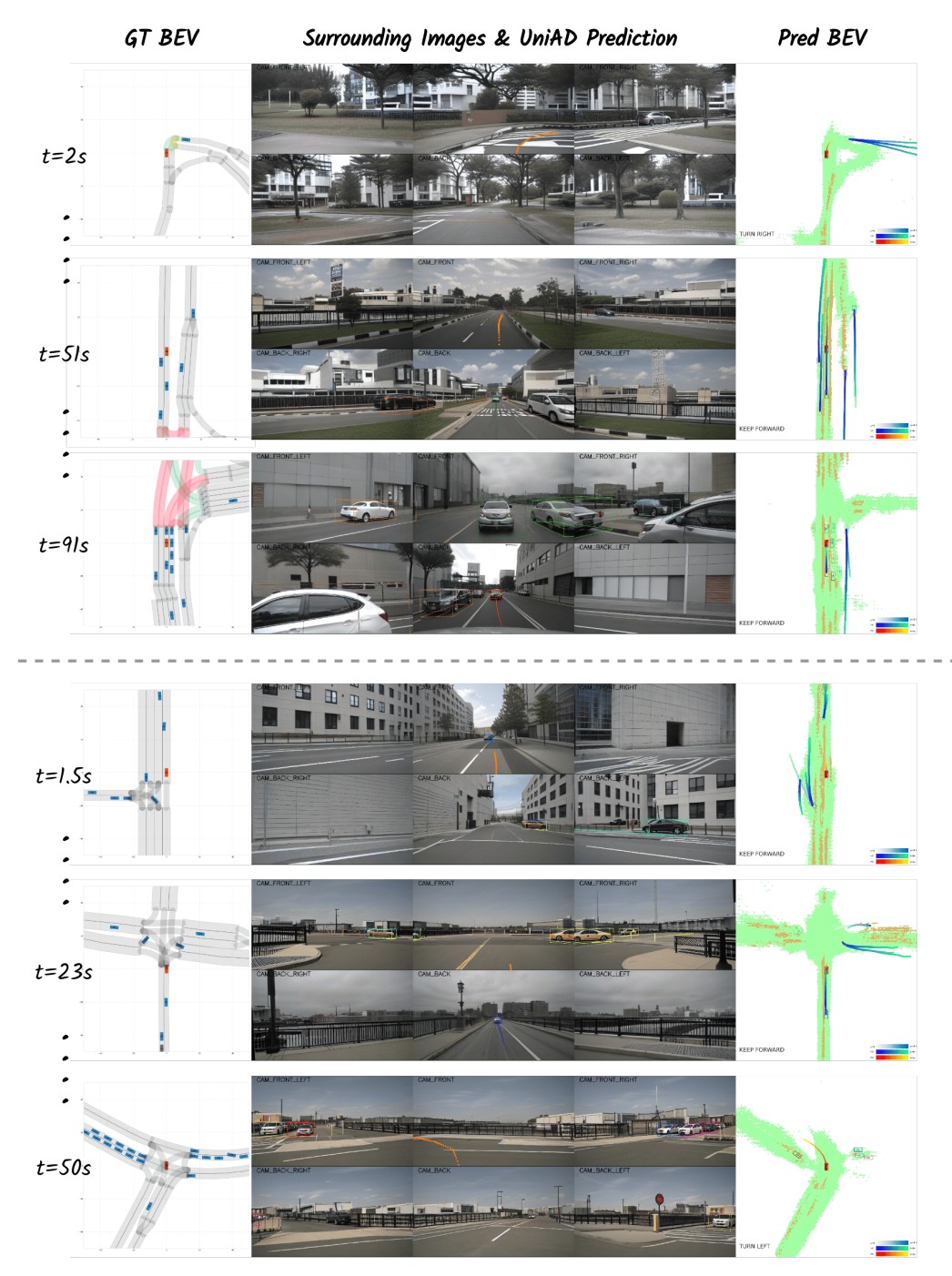

Figure 12: Case studies of UniAD's open-loop performance in DRIVEARENA. The figure presents two long-term open-loop simulation sequences: the upper sequence depicts a Singapore road network and style (left-hand drive), while the lower sequence shows a Boston road network and style (right-hand drive). Each subfigure displays, from left to right: Traffic Manager's ground truth BEV; World Dreamer-generated image with corresponding UniAD detection bounding boxes and predicted trajectories; and UniAD-predicted BEV image.

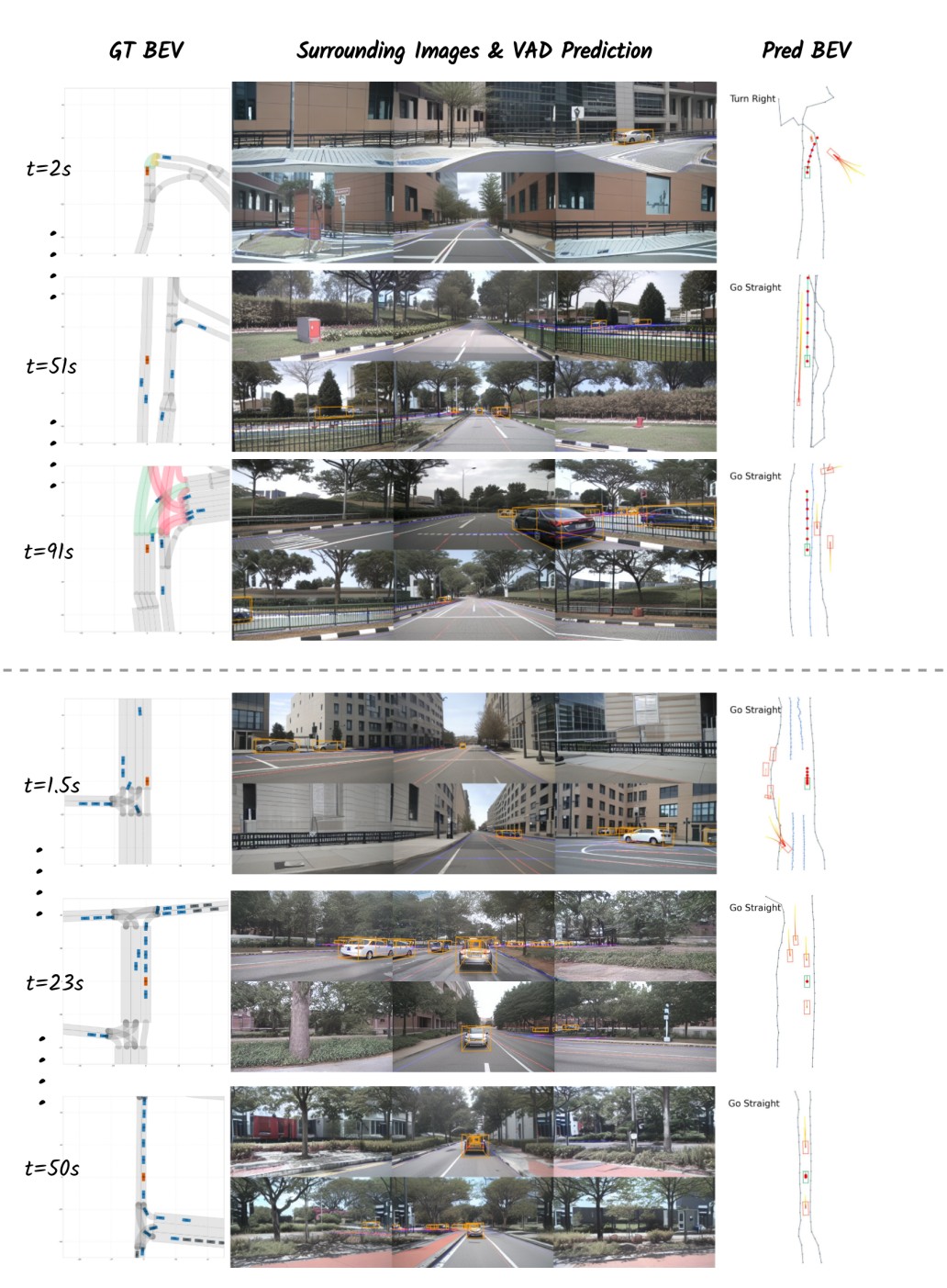

Figure 13: Examples of VAD's open-loop performance in DRIVEARENA. The figure presents two long simulation sequences: the upper one captures a Singapore road network with left-hand driving, while the lower one shows a Boston road network with right-hand driving. Each subfigure provides, from left to right: the ground truth BEV from Traffic Manager; images generated by World Dreamer with ground truth layout; and VAD's predicted BEV representation.

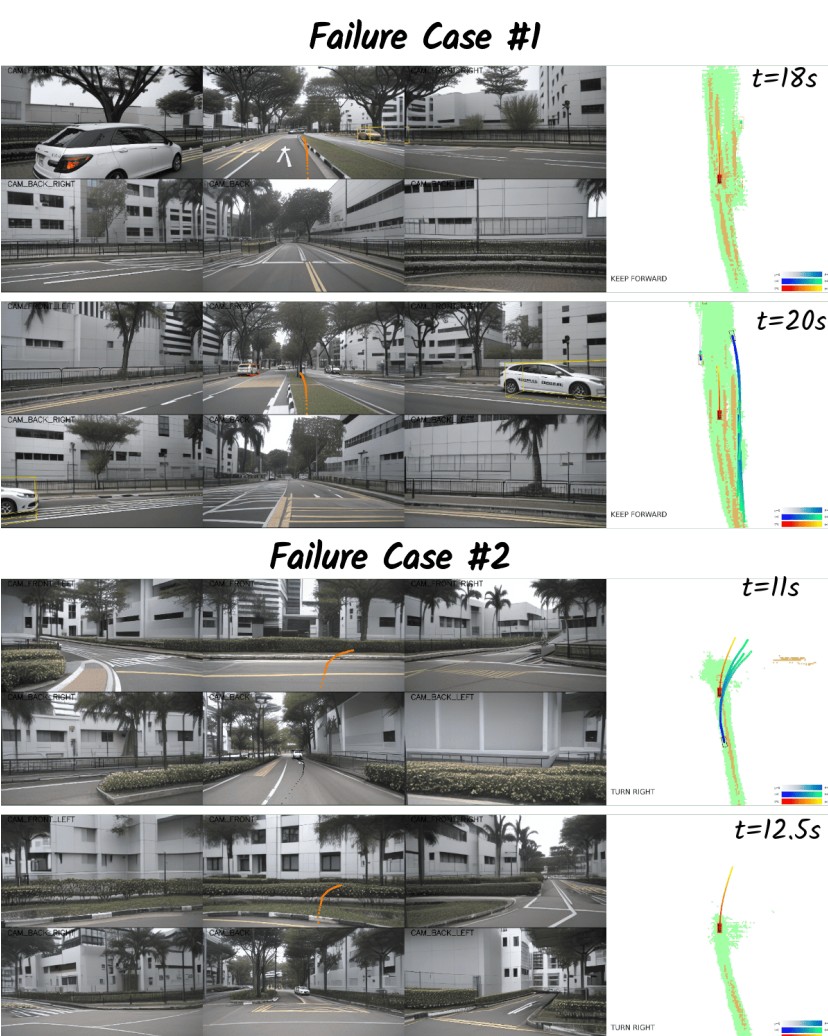

Figure 14: Failure cases of UniAD in DRIVEARENA's closed-loop mode. While UniAD generally predicts road structures accurately: (top) UniAD encroaching onto the central median; (bottom) UniAD failing to complete a right turn successfully.

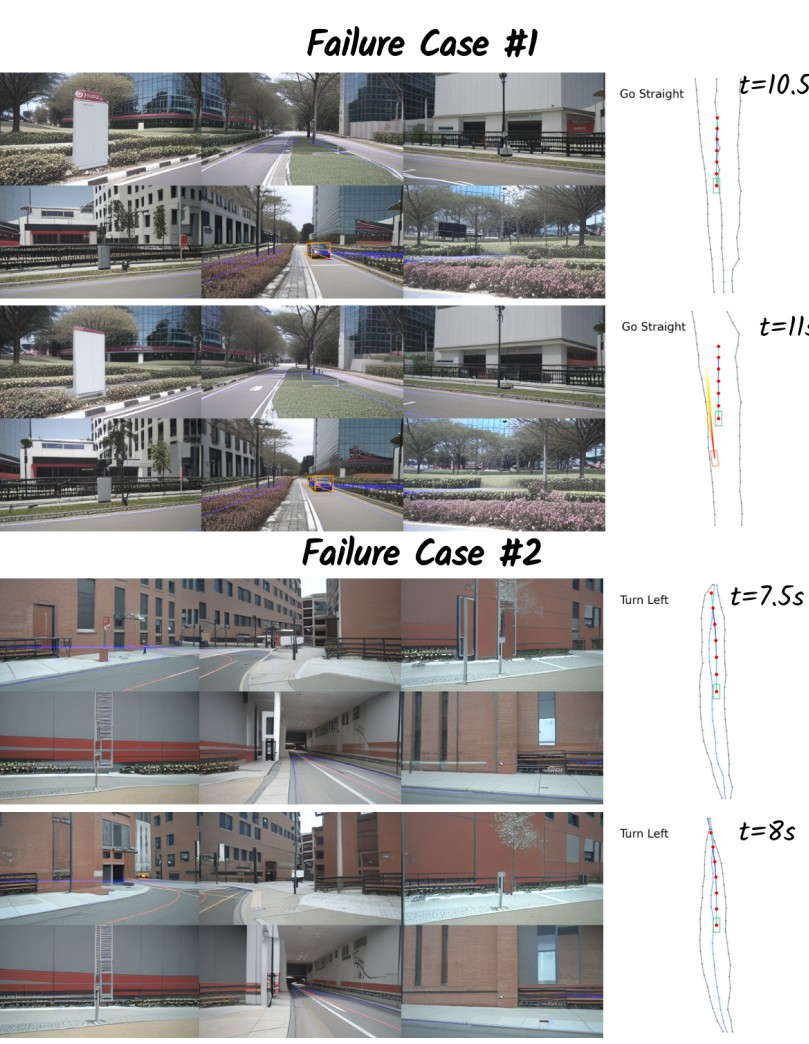

Figure 15: Examples of VAD failures in DRIVEARENA closed-loop mode. (Top) Although VAD was able to predict the road structure with basic accuracy, it drove onto the center greenbelt; (Bottom) VAD was unable to predict the left-turning road structure and therefore was unable to successfully complete the left turn.

## A.5 CORNER CASE GENERATION THROUGH INTSIM

In this section, we demonstrate one application of DRIVEARENA: generating extreme case or accident scene replays. Specifically, we utilize an algorithm called IntSIM to simulate accident traffic flow, where an attacking vehicle intentionally collides with others. This simulation reveals rare and extreme scenarios, providing valuable insights for researchers and engineers to test and improve the safety features of driving agent algorithms. Furthermore, DRIVEARENA allows various perspectives to be adopted during the simulation, so that the ego vehicle can be the vehicle affected by the collision, the vehicle causing the collision, or an observer witnessing the event.

Figure 16 illustrates a collision simulation within DRIVEARENA, showing two scenarios in which traffic participants attack the ego vehicle. As shown in the figure, DRIVEARENA is able to effectively simulate these traffic flows and render realistic surround images. However, since World Dreamer is trained entirely on the nuScenes dataset, which lacks extreme and unsafe car accident event data, it is temporarily unable to simulate the post-crash state of the vehicle or the physical impact of the collision, which is also a direction worth exploring further.

### Collision #1

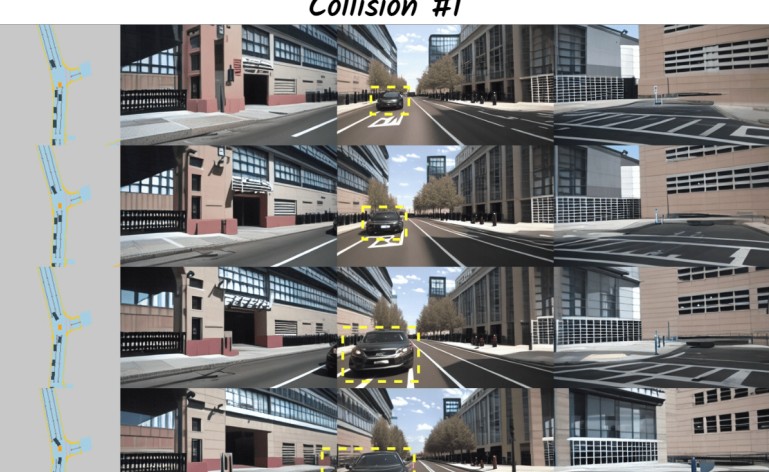

### Collision #2

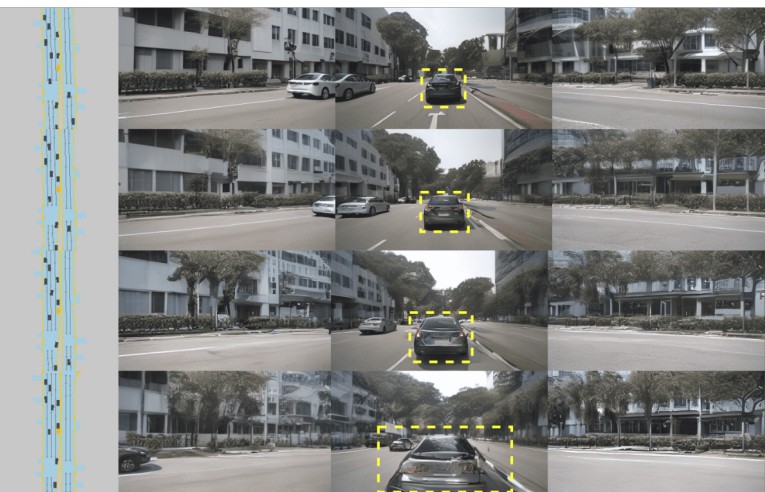

Figure 16: Simulated collision demonstrations within DRIVEARENA. Two extreme scenarios are shown in which one of the traffic participants initiates an attack on the ego vehicle while driving. The results show that the DRIVEARENA can handle these rare scenarios effectively. However, due to the limited availability of data on unsafe events, World Dreamer is unable to simulate the resulting vehicle damage or the physical impact of collisions.

## A.6 FUTURE WORK

In future work, the following limitations of the current DRIVEARENA implementation need to be addressed to improve its overall performance and capabilities:

**1) Data Diversity:** The current generative model is trained solely on the nuScenes dataset, which limits the diversity and emergence capabilities. We plan to expand training to include more varied datasets to enhance the model's robustness and versatility.

**2) Temporal Consistency:** While we can generate continuous videos with an autoregression strategy, maintaining motion trends and temporal consistency between frames remains challenging. Future work will explore multi-frame autoregressive networks and more scalable architectures (Peebles & Xie, 2023) to address these issues.

**3) Runtime Efficiency:** Like many generative models, World Dreamer requires significant runtime. Investigating faster sampling methods (Lu et al., 2022) and model quantization may alleviate these problems.

**4) Expanded Agent Testing:** We plan to incorporate a broader range of driving agents within DRIVEARENA, facilitating the continuous learning and evolution of knowledge-driven driving agents in the closed-loop environment (Li et al., 2023c).

**5) A Real Arena:** DRIVEARENA can not only evaluate the performance of different driving agents, but also act as a testing ground for AD generative models. Using the same driving agent as a referee can fairly assess the sim-to-real gap of different generative models. This approach even provides a more credible and convincing evaluation compared to traditional metrics like FID and FVD.

We recognize that practical application may still be a way off, but the potential and promise shown by this work are evident. We hope this research will advance closed-loop exploration in highly realistic environments and offer a valuable platform for developing and assessing driving agents across a range of challenging scenarios. We encourage the community to collaborate in advancing this field. The era of open loops is transitioning, and autonomous driving evaluation and learning are set to enter a new era of closed-loop systems.

