# OpenReview forum: "DriveArena: A Closed-loop Generative Simulation Platform for Autonomous Driving"
_ICLR.cc/2025/Conference — Submitted to ICLR 2025_

### Official Review · Reviewer_i9PT · 2024-10-30

**Soundness:** 3
**Presentation:** 3
**Contribution:** 3
**Rating:** 6
**Confidence:** 4

**Summary:**

The paper proposes a traffic simulation platform for testing autonomous driving algorithms. The simulation architecture is built based on LimSim, using Monte Carlo tree search for vehicle motion planning. A diffusion-based renderer is applied to achieve realistic and controllable surround images for vision-based driving agents. The platform supports driving scenario construction from nuScenes and OSM; codes and tools are open-source for the community to use.

**Strengths:**

Compared to previous work using video generation methods as world models to achieve realistic traffic simulation, this work uses a two-step pipeline, including rule-based traffic simulation in geometric space and diffusion-based video generation conditioned on trajectory layouts of vehicles. I believe this approach can achieve better physical realism and temporal consistency.

Text prompts are introduced to achieve diverse driving scenarios and plenty of demos are presented to clearly show the generation results.

The codes are well organized with a modularized design and the whole platform is open-source to better support downstream tasks in research of autonomous driving.

**Weaknesses:**

The author says that one of the main contributions is scalability, which means simulation on any region can be achieved by using map info from OSM. As far as I know, OSM only contains road-level map information, and extra efforts like completing lane topology and planning vehicle OD are needed to construct simulations based on it, this part of the work seems unclear in this paper.

As the dreamer is the most important part of this paper, it would be better if the author could provide some indicators that can directly evaluate the generated results, like FID.

Minor note: it seems that Figure 3 is not in vector format.

**Questions:**

+ About constructing traffic scenarios from OSM.
  + How is the HD map built from the OSM data, and how is traffic demand generated in this kind of scenario?
  + OSM maps are not high quality in many areas, is there a way to solve this?
  + Is this part of the work mainly based on the tools provided by SUMO or LimSim?
+ Are there any indicators to evaluate the generated results directly, like FID or FVD?

---

> ### Author Response · Authors · 2024-11-19
> **Author Response for Reviewer i9PT**
>
> Dear Reviewer i9PT:
>
> Thank you for your acknowledgement for our approach and constructive comments. We provide discussions and explanations about your concerns as follows.
>
> **Q1: About constructing traffic scenarios from OSM.**
>
> **A1:** Thanks for your question. Let me elaborate on how DriveArena utilizes OSM maps and generates traffic scenarios.
>
> While OSM only contains road-level map information, we employ a two-stage process using SUMO tools for map processing. Initially, we utilize the *OSMWebWizard* tool to download OSM maps and establish a topological roadnet. Subsequently, we employ the *randomTrips* script to create vehicle demands and their origin-destination pairs within the map. These steps constitute the pre-simulation process. During actual simulation, DriveArena's traffic manager module, which is a modified version of LimSim, takes control and manages background vehicle trajectory planning and interactions with the ego vehicle, ensuring vehicles reach their destinations according to the generated traffic demands.
>
> We acknowledge the concern about OSM's varying quality across different regions. To address this limitation, DriveArena supports simulation on any SUMO format representation maps. This flexibility provides users with multiple options: they can modify downloaded OSM maps according to their specific requirements, manually create custom maps, or convert OpenDrive format maps into the supported format. This approach offers users considerable freedom in creating diverse map types suitable for their specific simulation needs.
>
> We have also included a detailed elaboration of this process in Appendix A.2.2 of our revised manuscript.
>
> **Q2: Are there any indicators to evaluate the generated results directly, like FID or FVD?**
>
> **A2:** To address your concerns, we have included FID metric comparisons below. Our DriveArena achieves a 16.03 FID, which outperforms MagicDrive’s 16.20.
>
> | Method | FID↓ |
> | --- | --- |
> | MagicDrive | 16.20 |
> | DriveArena | 16.03 |
>
> However, we want to note that FID might not be the ideal metric for evaluation. Some recent works have pointed out that FID combines sample quality and diversity into a single value, making it unable to distinguish between these two important aspects [1] and lacking interpretability [2]. We believe using autonomous driving algorithms for fidelity evaluation is more intuitive and interpretable.
>
> **Q3: It seems that Figure 3 is not in vector format.**
>
> **A3:** Thank you for your careful observation! We apologize for affecting your reading experience. We have replaced this image with a better version in the revised version.
>
> [1] Kynkäänniemi, Tuomas, et al. "Improved precision and recall metric for assessing generative models." *Advances in neural information processing systems* 32 (2019).
>
> [2] Naeem, Muhammad Ferjad, et al. "Reliable fidelity and diversity metrics for generative models." *International Conference on Machine Learning*. PMLR, 2020.

---

> ### Author Response · Authors · 2024-11-25
>
> Dear Reviewer i9PT,
>
> We sincerely appreciate your time and effort in reviewing our manuscript and offering valuable suggestions .
>
> **As the author-reviewer discussion period is approaching its end, and given there will not be a second round of discussions,  we would like to confirm whether our responses have effectively addressed your concerns.**
>
> If you require further clarification or have any additional concerns, we remain fully committed to addressing them promptly.
>
> Best regards,
>
> Authors of Submission 2783

---

> > ### Comment · Reviewer_i9PT · 2024-11-27
> >
> > Thank you for your reply! I don't have any other questions.

---

### Official Review · Reviewer_UuYZ · 2024-11-01

**Soundness:** 3
**Presentation:** 3
**Contribution:** 3
**Rating:** 8
**Confidence:** 4

**Summary:**

This work presents DriveArena, an image-based high-fidelity simulator for closed-loop simulation of agents for autonomous driving. DriveArena consists of a Traffic Manager and a World Dreamer, and its modular design allows for easy replacement of each of the components. Traffic Manager enables the dynamic control of all traffic agents and supports various HD maps, both of which are inputs to World Dreamer, which uses conditional diffusion to generate realistic images. The diffusion model is conditioned on map and object layout and generates images autoregressively for temporal consistency and variable length simulations.

DriveArena is evaluated in two ways. First, its fidelity is evaluated, among others, with UniAD's performance. Then, open- and closed-loop evaluation of VAD and UniAD models is performed.

**Strengths:**

* Highly relevant research direction: The work correctly argues for the need of closed-loop evaluation of autonomous driving behavior models.
* Novelty: High-fidelity closed-loop image-based simulation with clear controllability (e.g. via text prompts).
* Performance: Evaluation of sim-to-real gap shows superiority over MagicDrive and reasonable results for open-loop and closed-loop behavior eval.
* Well presented: The paper is easy to follow and all information is well presented.

**Weaknesses:**

* Evaluation of sim-to-real gap: The presented evaluation is pretty short and for example lower L2 distances do not necessarily imply higher quality images. Additional evaluation of the fidelity would be helpful. Are there perception metrics such as FID that could be used or other metrics that compare statistics between ground truth and generated images? Otherwise, user studies are another possibility to judge the quality.
* Unclear takeaway from VAD vs. UniAD open- and closed-loop comparison: In open-loop, UniAD performs better on nuScenes but worse on DriveArena than VAD. This difference is explained with better open-loop generalization of VAD. However, it's unclear what role the fidelity of DriveArena plays. Is it possible to e.g. run an experiment with different DriveArena datasets, some that are closer and some that are further from nuScenes? In closed-loop eval, UniAD outperforms VAD in DriveArena. It's unclear whether these differences are due to open- / closed-loop model gaps or issues in DriveArena. I acknowledge that this difficulty of correct attribution is inherent to research in this area but you might be able to shed more light on this. For example, would it be possible to evaluate the models on various levels of fidelity in DriveArena to disentangle open- / closed-loop eval from it?

**Questions:**

* Evaluation of sim-to-real gap: How are the videos generated? Is this in open- or closed-loop? For correctness, it should be closed-loop. If so, do you notice any suffering from DAgger issues?

---

> ### Author Response · Authors · 2024-11-19
> **Author Response for Reviewer UuYZ (Part 1)**
>
> Dear Reviewer UuYZ:
>
> Thank you for your acknowledgment and constructive comments. We provide discussions and explanations about your concerns as follows.
>
> **Q1: Regarding the evaluation of sim-to-real gap: Additional evaluation of the fidelity would be helpful. Could perception metrics like FID be used between ground truth and generated images?**
>
> **A1:** To address your concerns, we have included FID metric comparisons below. However, we want to note that FID might not be the ideal metric for evaluation. Some works have pointed out that FID combines sample quality and diversity into a single value, making it unable to distinguish between these two important aspects [1] and lacking interpretability[2].
>
> | Method | FID↓ |
> | --- | --- |
> | MagicDrive | 16.20 |
> | DriveArena | 16.03 |
>
> We believe using driving agents for fidelity evaluation is more intuitive and interpretable. In Tables 1 and 2 of the manuscript, we present various metrics including 3D object detection, map segmentation, and other planning metrics to measure the quality of generated images. It's worth noting that in Table 1, due to computational resource constraints, both DriveArena and MagicDrive generate single images at 224×400 resolution, which are then upsampled 4x before being input to UniAD for inference. This inevitably introduces some performance loss.
>
> For comparison, we added a row in Table 1 showing results when the original nuScenes images are downsampled by 4x and then upsampled back to the original resolution. As shown in the second row of the table below, there is also some performance degradation compared to the original nuScenes dataset. When the resolution of generated images increases, these perception metrics are expected to improve[3].
>
> | Data Source | 3DOD |  | BEV Segmentation mIoU (%) |  |  |  | L2 (m)↓ |  |  |  | Col. Rate (%)↓ |  |  |  |
> | --- | --- | --- | --- | --- | --- | --- | --- | --- | --- | --- | --- | --- | --- | --- |
> |  | mAP↑ | NDS↑ | Lanes↑ | Drivable↑ | Divider↑ | Crossing↑ | 1.0s | 2.0s | 3.0s | Avg. | 1.0s | 2.0s | 3.0s | Avg. |
> | ori nuScenes | 37.98 | 49.85 | 31.31 | 69.14 | 25.93 | 14.36 | 0.51 | 0.98 | 1.65 | 1.05 | 0.10 | 0.15 | 0.61 | 0.29 |
> | **nuScenes w/ downsample** | 31.20 | 45.22 | 29.19 | 65.83 | 23.51 | 12.99 | 0.60 | 1.10 | 1.85 | 1.18 | 0.08 | 0.28 | 0.66 | 0.34 |
> | MagicDrive | 12.92 | 28.36 | 21.95 | 51.46 | 17.10 | 5.25 | 0.57 | 1.14 | 1.95 | 1.22 | 0.10 | 0.25 | 0.70 | 0.35 |
> | DRIVEARENA | 16.06 | 30.03 | 26.14 | 59.37 | 20.79 | 8.92 | 0.56 | 1.10 | 1.89 | 1.18 | 0.02 | 0.18 | 0.53 | 0.24 |
>
> [1] Kynkäänniemi, Tuomas, et al. "Improved precision and recall metric for assessing generative models." *Advances in neural information processing systems* 32 (2019).
>
> [2]  Naeem, Muhammad Ferjad, et al. "Reliable fidelity and diversity metrics for generative models." *International Conference on Machine Learning*. PMLR, 2020.
>
> [3] Gao, Ruiyuan, et al. "Magicdrive: Street view generation with diverse 3d geometry control." *arXiv preprint arXiv:2310.02601* (2023).

---

> > ### Comment · Reviewer_UuYZ · 2024-11-21
> >
> > Thank you for the reply and additional data points. I totally agree that FID is not an ideal metric. My main concern was that the metrics provided in Table 1 were produced by chaining several ML models, which makes attribution difficult, particularly when using driving agents, which might also suffer from distribution shifts.
> >
> > Your idea behind downsampling and upsampling nuScenes and adding it to Table 1 is great. The results clearly indicate the correlation between data quality and your eval metrics (except for L2 at 1s), which helps build trust in your fidelity evaluation!

---

> ### Author Response · Authors · 2024-11-19
> **Author Response for Reviewer UuYZ (Part 2)**
>
> **Q2: It's unclear what role the fidelity of DriveArena plays.  In closed-loop eval, UniAD outperforms VAD in DriveArena.  It's unclear whether these differences are due to open- / closed-loop model gaps or issues in DriveArena.**
>
> **A2:** This is a great question. Let’s try to discuss this comprehensively.
>
> 1. By comparing the first and second rows in Table 2, we observe that both VAD and UniAD models show minimal performance differences between the original and generated datasets. This suggests that from the driving agent's perspective,  the images generated by World Dreamer in *original nuScenes scenarios* maintain high similarity with the original nuScenes images.
> 2. Through careful analysis, we believe there exists a domain bias between DriveArena and nuScenes. This bias manifests in two key aspects: First, the image quality and fidelity of World Dreamer-generated images still have room for improvement, particularly in terms of temporal consistency. Second, the traffic flow patterns and vehicle interaction behaviors simulated by the Traffic Manager module show certain differences from the original nuScenes dataset in key characteristics such as traffic density. We intentionally maintain these differences to test the generalization capabilities of driving agents.
> 3. In DriveArena's open-loop mode, VAD indeed achieves higher PDMS metrics than UniAD (contrary to nuScenes open-loop results), but this is primarily due to VAD scoring nearly 20% higher in trajectory comfort metrics (C). For other critical metrics like collision avoidance and drivable area compliance, both models perform similarly.
>
>     However, in closed-loop mode, UniAD outperforms VAD in driving score metrics, mainly because UniAD can drive much longer route, thus achieving higher Route Completion (RC) rates.
>
>
> In conclusion, we believe that by using DriveArena's comprehensive evaluation standards(both open-loop and closed-loop metrics), we can minimize the impact of sim-to-real gaps and better reflect the inherent capability differences between AD models.
>
> **Q3: Would it be possible to evaluate the models on various levels of fidelity in DriveArena to disentangle open- / closed-loop eval from it?**
>
> **A3**: Following your suggestion, we conducted additional evaluations in DriveArena using models with different fidelity levels. Specifically, we employed an internally developed World Dreamer generation model with enhanced realism and better temporal consistency, which demonstrates improved performance with a reduced FID score from 16.03 to 14.6. We tested the new model in both open-loop and closed-loop modes, and the results are shown in the following table. (`DriveArena*` denotes the framework integrated with our improved World Dreamer model)
>
> | Scenario | Driving Agent | NC ↑ | DAC ↑ | EP ↑ | TTC ↑ | C ↑ | PDMS ↑ |
> | --- | --- | --- | --- | --- | --- | --- | --- |
> | DriveArena | VAD | 0.807±0.11 | 0.950±0.05 | 0.795±0.13 | 0.800±0.12 | 0.913±0.09 | 0.683±0.12 |
> |  | UniAD | 0.792±0.11 | 0.942±0.04 | 0.738±0.11 | 0.771±0.12 | 0.749±0.16 | 0.636±0.08 |
> | DriveArena*  | VAD | 0.829±0.08 | 0.954±0.05 | 0.767±0.07 | 0.815±0.11 | 0.920±0.10 | 0.687±0.05 (+0.004) |
> |  | UniAD | 0.843±0.04 | 0.958±0.05 | 0.728±0.06 | 0.829±0.05 | 0.704±0.14 | 0.669±0.02 (+0.033) |
> | nuScenes GT | Human | 1.000±0.00 | 1.000±0.00 | 1.000±0.00 | 0.979±0.12 | 0.752±0.17 | 0.950±0.06 |
>
> | Driving Agent | Sim | PDMS ↑ | ADS ↑ |
> | --- | --- | --- | --- |
> | VAD in `bos_route_1` | DriveArena | 0.5830 | 0.0352 |
> |  | DriveArena*  | 0.6140 (+0.0310) | 0.0532 |
> | UniAD in `bos_route_1` | DriveArena | 0.4952 | 0.0450 |
> |  | DriveArena* | 0.7401 (+0.2449) | 0.0760 |
>
> **Open-loop**: When using generated images from a more fidelity World Dreamer, UniAD showed a 5% improvement in open-loop PDMS metrics,  while VAD  only achieved a 0.5% improvement.
>
> **Closed-loop**: More notably, in the closed-loop evaluation on Boston-route 1, UniAD demonstrated a remarkable 50% enhancement in PDMS metrics, while VAD only showed a modest 5% improvement.
>
> These results clearly demonstrate that **the** **improved fidelity of DriveArena directly translates into enhanced driving agent performance**. Moreover, the driving agent exhibits **consistent behavior across WorldDreamer implementations** of varying fidelity levels. This underscores the practical value and effectiveness of our work.

---

> > ### Comment · Reviewer_UuYZ · 2024-11-21
> >
> > Thank you for the additional experiments! It's great to see that the higher fidelity DriveArena* leads to better open-loop and closed-loop eval metrics!
> >
> > Do you have an explanation for why UniAD closed-loop PDMS score in DriveArena* improved by 50%, while VAD's scores only improved by 5%?

---

> > > ### Author Response · Authors · 2024-11-22
> > >
> > > Dear reviewer UuYZ:
> > >
> > > Thank you for your timely response. We are pleased that our additional experiments have effectively addressed your concerns.
> > >
> > > Regarding the observation that *UniAD demonstrates greater improvements than VAD in the DriveArena\* closed-loop experiment*, we would like to offer our preliminary analysis. As evidenced in Table 1 of Response Part 2, when the World Dreamer's quality improves, UniAD exhibits substantial gains on most metrics, even surpassing VAD's performance indicators. Similarly, Table 2 reveals an even more pronounced enhancement in UniAD's metrics. These two pieces of evidence suggest that **UniAD possesses higher sensitivity to temporal consistency in generated image sequences**, allowing it to achieve more significant improvements when presented with temporally coherent video sequences.
> > >
> > > We sincerely appreciate your careful review and feedback, which have helped strengthen our manuscript. Your support and recognition is greatly valued.

---

> > > > ### Comment · Reviewer_UuYZ · 2024-11-26
> > > >
> > > > Thank you for your answer as well as your previous answers and updates. I have updated the rating accordingly!

---

> ### Author Response · Authors · 2024-11-19
> **Author Response for Reviewer UuYZ (Part 3)**
>
> **Q4: Evaluation of sim-to-real gap: How are the videos generated? Is this in open- or closed-loop?**
>
> **A4:** World Dreamer generates images solely based on the input scene layout information, including map layouts and object bounding boxes, without distinguishing whether these layout inputs originate from open-loop or closed-loop simulation.
>
> On our provided webpage, unless specifically noted otherwise, the videos predominantly showcase open-loop driving scenarios. We also demonstrated some closed-loop driving simulations. These open-loop visualization cases stem from a practical limitation: both UniAD and VAD struggle to maintain stable and safe driving in closed-loop environments for extended periods, which prevents us from fully demonstrating DriveArena's capabilities.
>
> **Q5: Do you notice any suffering from DAgger issues?**
>
> **A5**: Regarding the "DAgger problem", we want to confirm whether you meant to ask about the problem that the driving agent learns on the open-loop nuScenes dataset and suffers from degraded generalization performance on unseen DriveArena scenes.
>
> If so, we indeed observed significant performance limitations when driving agents are trained solely on open-loop datasets and deployed in closed-loop scenarios. As we highlighted in our introduction, existing driving datasets predominantly contain trajectory samples from straightforward driving scenarios, where *agents can achieve reasonable performance by simply maintaining their current speed.* More critically, in open-loop driving environments, each decision made by the agent is based on a relatively safe state. And *when the agent deviates from these safe trajectories, it often struggles to effectively correct its path*.
>
> This observation aligns with the fundamental challenge in AD: **agents trained on open-loop data lack the ability to recover from error states**, as they have never encountered such scenarios during training. This limitation is *precisely* one of our motivations for developing a closed-loop high-fidelity simulator, as it allows agents to learn and adapt to a broader range of driving scenarios, including recovery from dangerous corner cases.

---

> > ### Comment · Reviewer_UuYZ · 2024-11-21
> >
> > Thank you for the answers! Yes, my question targeted the data distribution shift between open-loop training data and closed-loop rollouts with respect to the agent "states" (particularly what you call "error states").

---

### Official Review · Reviewer_JafA · 2024-11-04

**Soundness:** 3
**Presentation:** 3
**Contribution:** 3
**Rating:** 6
**Confidence:** 5

**Summary:**

This work presents a generative simulation framework for autonomous driving. In particular, a layout-conditional diffusion model is proposed as a sensor simulator, with bounding boxes and road graphs serve as underlying world state.

**Strengths:**

The topic of generative simulation is an especially timely one for the AV field, with better and better generative video models coming out regularly and frameworks such as this one being able to capitalize on these parallel investments.

While all of the individual pieces of this framework have existed before, connecting them all together into a usable simulation framework for the broader community to use is appreciated and presents a potential path towards practical generative simulation.

The paper is written well and it is easy to follow the core ideas as presented.

Leveraging layout-conditional generation is a sound and sensible idea for maintaining consistency across time.

**Weaknesses:**

The core weakness is that any geometric or semantic consistency is not guaranteed. Layout conditioning certainly helps, but in the linked webpage's videos there are clear inconsistencies across timesteps (e.g., car color and types changing over time). This is something that is not brought up in Lines 190 - 198, but perhaps should be as it is the core reason why works leverage NeRF or 3D Gaussian Splatting (for their geometric/semantic/temporal consistency over purely-2D generative models).

While static images of generations appear to be of good quality, there are significant temporal consistency issues when viewed as part of a video on the linked project page (most videos appear to be static even with the ego-vehicle theoretically moving forward in the world). Do the authors have any idea for why that is? It almost appears that the video diffusion model suffers from mode collapse when tasked with generating building walls (taking example from the linked webpage).

The AV results in Tables 1 and 2 still show a significant gap to real data, indicating that, while the core points of DriveArena are sensible, there is still much work to be done to leverage it for practical AV development.

**Questions:**

In Lines 261-263 it is written "We also verify that extending to a multi-frame auto-regressive version (using multiple past frames as reference and outputting multi-frame images) and adding additional temporal modules can enhance temporal consistency." - How does this work in practice? In closed-loop, the ego-vehicle can drive however it wants and so there can still be inconsistencies between generated videos at time t and t+1, right? Further, how are the generated videos used? Are T frames predicted but only the 1st one is shown to the ego-policy? And then a new prediction is made (similar to model-predictive control)?

---

> ### Author Response · Authors · 2024-11-19
> **Author Response for Reviewer JafA (Part 1)**
>
> Dear Reviewer JafA:
>
> Thank you for your acknowledgment of DriveArena’s idea and constructive comments. We provide the following discussions and explanations regarding your concerns.
>
> **Q1: The core concerns raised involve geometric/semantic consistency issues and temporal coherence problems in the generated videos.**
>
> **A1:** Let's discuss geometric/semantic consistency and temporal coherence issues separately.
>
> 1. Geometric/semantic inconsistency:
>
>     We acknowledge that World Dreamer lacks geometric and semantic consistency guarantees due to the absence of a 3D model constraint.
>
>     While some recent studies have attempted to reconstruct 3D scenes from long generated video sequences (e.g., MagicDrive3D[1] and DriveDreamer4D[2]), their reconstruction results are based on generated videos—a capability that World Dreamer also possesses. Using the generated model to help improve the performance of the reconstructed model is also a direction worth exploring, but it is beyond the scope of our current discussion. We have supplemented the above issues in the updated version of the manuscript (line 198-199).
>
>     Our approach mitigates these geometric inconsistencies by introducing scene layout conditional constraints, which effectively maintain consistency in road topology and vehicle positioning. As demonstrated in Figure 6 of our manuscript, driving agents can successfully interpret our generated images and accurately extract map and vehicle information.
>
> 2. Temporal inconsistency :
>
>     We acknowledge certain limitations in DriveArena, which operates on a single-frame input-output basis. However, we are actively developing a temporal version (as shown in our project page demonstration: https://blindpaper.github.io/DriveArena/#Infinite_Multi-View_Video_Generation) that has already shown promising results. The academic community has made significant breakthroughs in temporal consistency through works like Panacea [3] and DrivingDiffusion[4], and we plan to integrate these advanced generative models into the DriveArena framework to enhance its temporal consistency.
>
>
> While improving scene continuity and consistency would undoubtedly enhance simulator performance, but: Is geometric/temporal consistency the only need for current AD technology? We believe that high-fidelity images and closed-loop interactivity are more crucial aspects of a simulator. Our DriveArena approach enables constructing more realistic corner case scenarios (as shown in Appendix A5), allowing for better evaluation of autonomous driving algorithms. This addresses some limitations of existing simulators like Carla and represents a promising research direction.
>
> **Q2: It appears that the video diffusion model suffers from mode collapse problems.**
>
> **A2**: This can be attributed to the limited training data from the nuScenes dataset, which only contains approximately *4 hours of video data*. Additionally, since World Dreamer employs an autoregressive generation approach, the model indeed exhibits some mode collapse behavior. To address this limitation, one potential solution could be to periodically incorporate style reference images during generation, which might help increase the diversity of the generated images.
>
> [1] Gao, Ruiyuan, et al. "MagicDrive3D: Controllable 3D Generation for Any-View Rendering in Street Scenes." *arXiv preprint arXiv:2405.14475* (2024).
>
> [2] Zhao, Guosheng, et al. "Drivedreamer4d: World models are effective data machines for 4d driving scene representation." *arXiv preprint arXiv:2410.13571* (2024).
>
> [3] Wen, Yuqing, et al. "Panacea: Panoramic and controllable video generation for autonomous driving." *Proceedings of the IEEE/CVF Conference on Computer Vision and Pattern Recognition*. 2024.
>
> [4] Li, Xiaofan, Yifu Zhang, and Xiaoqing Ye. "DrivingDiffusion: Layout-Guided multi-view driving scene video generation with latent diffusion model." *arXiv preprint arXiv:2310.07771* (2023).

---

> ### Author Response · Authors · 2024-11-19
> **Author Response for Reviewer JafA (Part 2)**
>
> **Q3: The AV results in Tables 1 and 2 still show a significant gap to real data, there is still much work to be done to leverage it for practical AV development.**
>
> **A3:** Compared to the MagicDrive baseline, DriveArena demonstrates significant improvements in metrics such as segmentation and agent collision rate. However, indeed, as shown in Tables 1 and 2, there is still a noticeable gap between DriveArena's generated images and nuScenes GT. As you pointed out, DriveArena, being the first generative controllable closed-loop simulation platform, still has room for improvement.
>
> For instance, due to current computational resource constraints, DriveArena generates images at a resolution of 224\*400, which is four times smaller than the original nuScenes images (900\*1600). This resolution difference significantly impacts various fidelity metrics. One of our future directions is to increase the generation resolution to further reduce the sim-to-real gap in DriveArena.
>
> **Q4: How does the multi-frame auto-regressive version of the diffusion model work in practice? Further, how are the generated videos used? Are T frames predicted but only the 1st one is shown to the ego-policy, and then a new prediction is made (similar to model-predictive control)?**
>
> **A4:** As mentioned in the paper, in the multi-frame version, we reference multiple past frames and output multi-frame images with additional temporal modules, which helps the diffusion model better capture the motion patterns between frames and generate videos with improved temporal consistency.
>
> For driving agents trained on nuScenes, i.e. UniAD, they follow a planning frequency of 2 Hz, while our Traffic Manager module operates at 10Hz. When the simulation starts, the temporal version of our generation model outputs at 10Hz, generating 7 frames each time, where the first 2 frames overlap with the last two frames from the previous output, resulting in 5 new generated frames. We then feed the last generated frame to the driving agent running at 2Hz for the next planning step. This approach ensures that the generated videos appear more continuous and smooth while maintaining proper closed-loop simulation.
>
> Preliminary results of the temporal version can be found at the end of our project website  https://blindpaper.github.io/DriveArena/#Infinite_Multi-View_Video_Generation. We hope this addresses your questions about the multi-frame World Dreamer version.

---

> ### Author Response · Authors · 2024-11-25
>
> Dear Reviewer JafA,
>
> We sincerely appreciate your time and effort in reviewing our manuscript and offering valuable suggestions .
>
> **As the author-reviewer discussion period is approaching its end, and given there will not be a second round of discussions,  we would like to confirm whether our responses have effectively addressed your concerns.**
>
> Should you require any additional clarification or have remaining questions, we remain fully committed to addressing them.
>
>
> Best regards,
>
> Authors of Submission 2783

---

> ### Author Response · Authors · 2024-12-02
>
> Dear Reviewer JafA,
>
> As we are now in the final day of the extended rebuttal period, we would greatly value any feedback on our previous responses to ensure we have addressed any of your concerns.
>
> **Even a brief confirmation would be immensely helpful for us.**
>
> While we fully understand your busy schedule, we would greatly appreciate any response you could provide during these final hours of the discussion period.
>
> Authors of Submission 2783

---

> > ### Comment · Reviewer_JafA · 2024-12-02
> > **Apologies for the delayed response**
> >
> > My apologies for the delay, the rebuttal has cleared up all the questions I had and I am happy to maintain my score (ideally ICLR would allow us to use ratings like 6.5 or 7, as I have no remaining questions and all the weaknesses are plainly stated/understood, but I also don't want to greatly increase my score to an 8).

---

> > > ### Author Response · Authors · 2024-12-02
> > >
> > > Thank you very much for your response and for confirming that our rebuttal has successfully addressed all your questions.
> > >
> > > We greatly appreciate your acknowledgment of our manuscript's merits and understand your position regarding the scoring granularity.

---

### Official Review · Reviewer_Pxgv · 2024-11-04

**Soundness:** 2
**Presentation:** 2
**Contribution:** 2
**Rating:** 3
**Confidence:** 4

**Summary:**

The submission introduces DriveArena, a high-fidelity closed-loop simulation platform designed for testing and developing autonomous driving agents in real-world scenarios. The platform consists of two main components: the Traffic Manager and the World Dreamer. The Traffic Manager is responsible for generating realistic traffic flow on any global street map, while the World Dreamer is a high-fidelity conditional generative model that creates infinite autoregressive simulations. DRIVEARENA enables the generation of diverse traffic scenarios with varying styles and allows driving agents that can process real-world images to navigate within its simulated environment.

**Strengths:**

1. This submission targets an important problem in the field of autonomous driving: how to properly evaluate the performance of end-to-end systems. The submission introduces a closed-loop evaluation method, which is more reflective of real-world driving conditions compared to open-loop evaluations.  It will be useful for practical applications.
2. The platform utilizes road networks from cities worldwide and allows for the generation of diverse traffic scenarios with varying styles, which is essential for training and evaluating driving agents across different driving environments.
3. The submission provides a clear and detailed explanation of the technical aspects of DriveArena. The figures, tables, and appendices enhance the understanding of the system's components and their interactions.

**Weaknesses:**

1. The submission lacks novelty. The proposed DriveArena is just a combination of several existing methods: LimSim for traffic simulation and condition generation; DriveDreamer/Vista for generating images from the conditions; NAVSIM and Carla for closed-loop evaluation.
2. The World Dreamer model is trained primarily on the nuScenes dataset, which may not capture diverse driving scenarios. To improve the model's generalizability, it would be beneficial to incorporate additional datasets that represent different geographical locations, driving cultures, and road conditions.
3. This submission fails to address an important issue for closed-loop evaluation: the model should be able to generate the same scene captured from different positions (similar to actual scenarios of driving differently in the same scene). No visualization was found addressing such an issue.
4. Experiments are not enough. The submission primarily focuses on the simulation platform itself rather than an in-depth evaluation of various driving agents within the platform. Expanding the experimental section to include a broader range of driving agents and more extensive testing can help provide a clearer picture of DRIVEARENA's capabilities and limitations.
5. Minor issues in writing and presentation. For example, The figures are not vectorized for zooming in and they are suggested to be replaced.

**Questions:**

Can authors provide visualizations of generating consistent scenes based on slightly different conditions (e.g., the ego car moves differently), which is a key aspect for closed-loop evaluation?

---

> ### Author Response · Authors · 2024-11-19
> **Author Response for Reviewer Pxgv (Part 1)**
>
> Dear Reviewer Pxgv：
>
> Thank you for your review and comments. We provide the following discussions and explanations regarding your concerns.
>
> **Q1:  The submission lacks novelty. The proposed DriveArena is just a combination of several existing methods.**
>
> **A1:** As acknowledged by other reviewers (e.g., Reviewer JafA: "*connecting them all together ... is appreciated and presents a potential path towards practical generative simulation*". Reviewer UuYZ: “*Novelty: High-fidelity closed-loop image-based simulation with clear controllability*”), we are the first pioneering work to apply generative models as AD simulators. We firmly believe that **closed-loop evaluation of driving agents in realistic street scenarios is both necessary and practically valuable**. Moreover, connecting these modules into one closed-loop simulation platform is not trivial. Our modular architecture enables DriveArena to be compatible with different Traffic Managers and World Dreamer methods for simulating various driving agents, which is particularly valuable for the autonomous driving community.
>
> Besides, our World Dreamer differs from the DriveDreamer[1] and Vista[2] you mentioned: While DriveDreamer can generate continuous video clips, it cannot guarantee coherence between segments. Vista, as a world model, lacks control over background vehicles. We believe that controllability and long-term temporal coherence are crucial elements for generative models within a simulator. In contrast, our World Dreamer module not only achieves precise control over vehicles in the scene but also enables theoretically infinite-length video generation through an autoregressive paradigm.
>
> **Q2: The World Dreamer model is trained primarily on the nuScenes dataset. To improve the model's generalizability, it would be beneficial to incorporate additional datasets.**
>
> **A2:** Among various autonomous driving datasets, our choice of nuScenes as the training dataset for World Dreamer was based on several key considerations:
>
> 1. Representativeness and Popularity: the nuScenes dataset is one of the most widely adopted autonomous driving datasets. With its comprehensive annotations, it has become a benchmark for numerous autonomous driving approaches and generative models.
> 2. Diversity: Unlike the nuPlan dataset which is *limited to sunny daytime scenarios,* nuScenes data collection actually spans multiple cities and weather/daylight conditions, providing necessary diversity.
> 3. 360-degree Surround View: nuScenes provides complete surround-view image data, making it superior to the Waymo dataset, which lacks rear views.
>
> We also demonstrated generalizability through zero-shot inference on the nuPlan dataset (as shown in Appendix Figure 9). The results indicate that despite different camera settings and road network structures, our method can directly adapt to nuPlan's road networks without any training.
>
> Additionally, following your suggestion,  we also **trained a new version of World Dreamer using both nuPlan and nuScenes datasets.** By incorporating more diverse driving data, we further enhanced the generative model's generalization capability, enabling it to generate street scenes from Las Vegas and Pittsburgh (please refer to Figure 11 in the revised manuscript).
>
>
> [1] Wang, Xiaofeng, et al. "Drivedreamer: Towards real-world-driven world models for autonomous driving." *arXiv preprint arXiv:2309.09777* (2023).
>
> [2] Gao, Shenyuan, et al. "Vista: A Generalizable Driving World Model with High Fidelity and Versatile Controllability." *arXiv preprint arXiv:2405.17398* (2024).

---

> ### Author Response · Authors · 2024-11-19
> **Author Response for Reviewer Pxgv (Part 2)**
>
> **Q3: The model should be able to generate the same scene captured from different positions (similar to actual scenarios of driving differently in the same scene). No visualization was found addressing such an issue.**
>
> **A3:** To address your concern, we have included a new visualization of the "same scene from different viewpoints” in the revised manuscript (Figure 8 in the appendix). As shown, the road network and traffic participants remain consistent across two scenes, with the ego vehicle position shifting from the leftmost lane to the middle lane. While there are minor variations in front vehicle colors and street backgrounds, World Dreamer successfully maintains spatial consistency in lane markings and surrounding vehicle positions while preserving similar street styles and building configurations.
>
> This capability is achieved because World Dreamer uses both lane lines and 3D bounding boxes from multi-view as control conditions, along with reference images for style guidance. This enables DriveArena to maintain similarity when generating images of the same scene from different positions. However, we acknowledge that since World Dreamer doesn't incorporate a 3D scenario model to constrain geometric consistency (which is only achievable with 3DGS and NeRF-like methods), diffusion-based models theoretically cannot guarantee "completely identical" visual sequences when capturing the same scene from different positions.
>
> **Q4: Experiments are not enough. The submission primarily focuses on the simulation platform itself rather than an in-depth evaluation of various driving agents within the platform.**
>
> **A4:** We respectfully disagree with the reviewer's assessment regarding insufficient experimentation. Our experimental section encompasses comprehensive evaluations of both World Dreamer performance (including Fidelity, Controllability, and Scalability) and driving agent open-loop and closed-loop experiments. Additionally, in the appendix, we have provided extensive supplementary materials showcasing both successful and failed simulation cases of driving agents within DriveArena, as well as experiments demonstrating DriveArena's capability to generate collision corner cases.
>
> The core of our work lies in validating the feasibility of a generative model-based closed-loop simulator. Through the integration of driving agents, we have successfully demonstrated DriveArena's ability to produce high-fidelity, interactive environments for agent evaluation. As a pioneering work, DriveArena has already supported two mainstream open-source end-to-end AD agents: UniAD and VAD, subjecting them to thorough open-loop and closed-loop evaluations. Furthermore, we have introduced PDMS and Arena Driving score metrics to comprehensively assess agent performance.
>
> Through DriveArena's modular design, we are committed to collaborating with the AD community to enhance the platform. Our goal is to incorporate a wider variety of autonomous driving agents in the future, establishing DriveArena as a “real arena” for autonomous driving algorithms and providing a standardized testing environment for assessment and comparison for both academia and industry.
>
> **Q5: Minor issues in writing and presentation. For example, The figures are not vectorized for zooming in and they are suggested to be replaced.**
>
> **A5:** Thank you for your careful observation regarding the figure quality. We apologize for affecting your reading experience. We have replaced Figure 3 with high-resolution vectorized versions in the revised manuscript PDF.

---

> ### Author Response · Authors · 2024-11-25
>
> Dear Reviewer Pxgv,
>
> We sincerely appreciate your time and effort in reviewing our manuscript.
>
> We have provided detailed responses to your concerns several days ago. **As the author-reviewer discussion period is approaching its end, and given there will not be a second round of discussions,  we would like to confirm whether our responses have effectively addressed your concerns.**
>
> Should you require any additional clarification or have remaining questions, we remain fully committed to addressing them.
>
>
> Best regards,
>
> Authors of Submission 2783

---

> ### Author Response · Authors · 2024-12-02
>
> Dear Reviewer Pxgv,
>
> **As we are now in the final day of the extended rebuttal period,** we would greatly value any feedback on our previous responses to ensure we have addressed any of your concerns.
>
> While we fully understand your busy schedule, we would greatly appreciate any response you could provide during these final hours of the discussion period.
>
> Authors of Submission 2783

---

### Meta-Review · Area_Chair_f4Qy · 2024-12-23

**Metareview:**

This work proposes a closed-loop simulator for autonomous driving. The simulation involves two components: a traffic simulation system that generates traffic flow simulations and a multi-view image generation system for creating images based on the generated traffic and text prompt. The topic of AV simulation is timely and important, and the present work is clearly written. However, it is limited, as the underlying design choices limit geometric and semantic consistency, which ultimately puts the usefulness and contribution of such a platform into question. Given that the authors position this work from an overall system perspective, I am weighing the lack of this consistency higher. The authors are correct in that consistency is not the only thing needed for meaningful AV simulation, but it does seem to be an important requirement for such a work to be broadly useful for the community.

**Additional Comments On Reviewer Discussion:**

The authors addressed most of the concerns in the discussion with the reviewers. While one of the reviewers (Pxgv) did not update their scores or comment on the responses provided by the authors, most of this reviewer's concerns seem to be addressed (with the exception of consistency, which has been raised by other reviewers as well).

---

### Decision · Program_Chairs · 2025-01-22

Reject